



# Impact of Initialized Land Surface Temperature and Snowpack on Subseasonal to Seasonal Prediction Project, Phase I (LS4P-I): Organization and Experimental design

Yongkang Xue[1], Tandong Yao[2], Aaron A. Boone[3], Ismaila Diallo[1], Ye Liu[1], Xubin Zeng[4], William K.-M. Lau[5], Shiori Sugimoto[6], Qi Tang[7], Xiaoduo Pan[2], Peter J. van Oevelen[8], Daniel Klocke[9], Myung-Seo Koo[10], Zhaohui Lin[11], Yuhei Takaya[12], Tomonori Sato[13], Constantin Ardilouze[3], Subodh K. Saha[14], Mei Zhao[15], Xin-Zhong Liang[5], Frederic Vitart[16], Xin Li[2], Ping Zhao[17], David Neelin[1], Weidong Guo[18], Miao Yu[19], Yun Qian[20], Samuel S. P. Shen[21], Yang Zhang[18], Kun Yang[22], Ruby Leung[20], Jing Yang[23], Yuan Qiu[11], Michael A. Brunke[4], Sin Chan Chou[24], Michael Ek[25], Tianyi Fan[23], Hong Guan[26], Hai Lin[27], Shunlin Liang[28], Stefano Materia[29], Tetsu Nakamura[13], Xin Qi[23], Retish Senan[16], Chunxiang Shi[30], Hailan Wang[26], Helin Wei[26], Shaocheng Xie[7], Haoran Xu[5], Hongliang Zhang[31], Yanling Zhan[11], Weiping Li[32], Xueli Shi[32], Paulo Nobre[24], Yi Qin[22], Jeff Dozier[33], Craig R. Ferguson[34], Gianpaolo Balsamo[16], Qing Bao[35], Jinming Feng[11], Jinkyu Hong[36], Songyou Hong[10], Huilin Huang[1], Duoying Ji[23], Zhenming Ji[37], Shichang Kang[38], Yanluan Lin[22], Weiguang Liu[39,19], Ryan Muncaster[27], Yan Pan[18], Daniele Peano[29], Patricia de Rosnay[16], Hiroshi G. Takahashi[40], Jianping Tang[18], Guiling Wang[39], Shuyu Wang[18], Weicai Wang[2], Xu Zhou[2], Yuejian Zhu[26]

[1]University of California – Los Angeles, CA 90095, USA

[2]Institute of Tibetan Plateau Research, Chinese Academy of Sciences, China

[3]CNRM, University of Toulouse, Météo-France, CNRS, Toulouse, France

[4]University of Arizona, Tucson, USA

[5]Earth System Science Interdisciplinary Center (ESSIC), University of Maryland, College Park, USA

[6]Japan Agency for Marine Earth Science and Technology (JAMSTEC), Japan

[7]Lawrence Livermore National Laboratory, Livermore, CA 94550, USA

[8]International GEWEX Project Office, George Mason University, USA

[9]Hans Ertel Centre for Weather Research, Germany

[10]Korea Institute of Atmospheric Prediction Systems, South Korea

[11]Institute of Atmospheric Physics, Chinese Academy of Sciences, China

[12]Meteorological Research Institute, Japan Meteorological Agency, Japan

[13]Hokkaido University, Japan

[14]Indian Institute of Tropical Meteorology, Ministry of Earth Sciences, India

[15]Bureau of Meteorology, Australia

[16]European Centre for Medium-range Weather Forecasts (ECMWF), UK

[17]Chinese Academy of Meteorological Sciences, China Meteorological Administration, China

[18]School of Atmospheric Sciences, Nanjing University, China

[19]Nanjing University of Information Science Technology, Nanjing 210044, China

[20]Pacific Northwest National Laboratory, Richland, WA 99352, USA

[21]San Diego State University, USA

[22]Tsinghua University, China





[23]Beijing Normal University, China

[24]National Institute for Space Research (INPE), Brazil

[25]National Center for Atmospheric Research (NCAR), USA

[26]National Center for Environmental Prediction (NCEP)/ National Weather Service/
National Oceanic and Atmospheric Administration (NOAA), USA

[27]Environment and Climate Change Canada, Canada

[28]University of Maryland, College Park, USA

[29]Euro-Mediterranean Centre on Climate Change Foundation (CMCC), Italy

[30]National Meteorological Information Center, China Meteorological Administration, China

[31]National Meteorology Center, China Meteorological Administration, China

[32]National Climate Center, China Meteorological Administration, China

[33]University of California, Santa Barbara, USA

[34]Atmospheric Sciences Research Center, University at Albany, State University of New York, Albany, NY, 12203

[35]State Key Laboratory of Numerical Modeling for Atmospheric Sciences and Geophysical Fluid Dynamics (LASG), Institute of Atmospheric Physics, Chinese Academy of Sciences, China

[36]Yonsei University, South Korea

[37]Sun Yat-Sen University, China

[38]Northwest Institute of Eco-environment and Resources, Chinese Academy of Sciences, China

[39]University of Connecticut, USA

[40]Tokyo Metropolitan University, Japan

*Correspondence to*: Yongkang Xue (yxue@geog.ucla.edu)



**Abstract**. Sub-seasonal to seasonal (S2S) prediction, especially the prediction of extreme hydroclimate events such as droughts and floods, is not only scientifically challenging but has

substantial societal impacts. Motivated by preliminary studies, the Global Energy and Water Exchanges (GEWEX)/Global Atmospheric System Study (GASS) has launched a new initiative called "Impact of initialized **L**and **S**urface temperature and **S**nowpack on **S**ub-seasonal to **S**easonal **P**rediction" (LS4P), as the first international grass-root effort to introduce spring land surface temperature (LST)/subsurface temperature (SUBT) anomalies over high mountain areas

as a crucial factor that can lead to significant improvement in precipitation prediction through the remote effects of land/atmosphere interactions. LS4P focuses on process understanding and predictability, hence it is different from, and complements, other international projects that focus on the operational S2S prediction. More than forty groups worldwide have participated in this effort, including 21 Earth System Models, 9 regional climate models, and 7 data groups.

This paper overviews the history and objectives of LS4P, provides the first phase experimental protocol (LS4P-I) which focuses on the remote effect of the Tibetan Plateau, discusses the LST/SUBT initialization, and presents the preliminary results. Multi-model ensemble experiments and analyses of observational data have revealed that the hydroclimatic effect of the spring LST in the Tibetan Plateau is not limited to the Yangtze River basin but may

have a significant large-scale impact on summer precipitation and its S2S prediction. LS4P models are unable to preserve the initialized LST anomalies in producing the observed anomalies largely for two main reasons: i) inadequacies in the land models arising from total soil depths which are too shallow and the use of simplified parameterizations which both tend to limit the soil memory; and ii) reanalysis data, that are used for initial conditions, have large discrepancies

from the observed mean state and anomalies of LST over the Tibetan Plateau. Innovative approaches have been developed to largely overcome these problems.



## 1. Introduction

Subseasonal-to-seasonal (S2S) prediction, especially the prediction of extreme hydroclimatic events such as droughts and floods, is not only scientifically challenging but also has substantial societal impacts since such phenomena can have serious agricultural, economic, and ecological consequences (Merryfield et al., 2020). However, the prediction skill for precipitation anomalies in spring and summer months, a significant component of extreme climate events, has remained stubbornly low for years. It is therefore important to understand the sources of such predictability and to develop more reliable monitoring and prediction capabilities. Various mechanisms have been attributed to the S2S predictability. For instance, oceanic basin-wide tropical sea surface temperature (SST) anomalies are known to play a major role in causing extreme events. The connection between SST [e.g., El Niño Southern Oscillation (ENSO), Pacific Decadal Oscillation (PDO), Atlantic Multidecadal Oscillation (AMO), and Madden–Julian oscillation (MJO)] and the associated weather and climate predictability has been extensively studied for decades (Trenberth et al., 1988; Ting and Wang, 1997; Barlow et al., 2001; Schubert et al., 2008; Jia and Yang, 2013; Seager et al., 2014). The linkage of extreme hydrological events to tropical ocean basin SST anomalies allows us to predict them with useful skill at long lead times, ranging from a few months to a few years. Despite significant correlations and demonstrated predictive value, numerous studies based on observational data analyses and numerical simulations have consistently shown that SST alone only partially explains the phenomena of predictability (Rajagopalan et al., 2000; Schubert et al., 2004, 2009; Scaife et al., 2009; Mo et al., 2009; Rui and Wang, 2011; Pu et al., 2016; Xue et al., 2016a, b, 2018). For instance, the 2015-2016 El Niño event, one of the strongest since 1950, was associated with an extraordinary Californian drought, while the 2016-2017 La Niña event has been related to record rainfall that effectively ended the 5-year Californian drought, contrary to established canonical SST-Californian drought/flood relationships. In South America, there is also an example where the canonical association of thermally direct, SST-driven atmospheric circulation fails (Robertson and Mechoso, 2000; Nobre et al., 2012). Although an important role for random atmospheric internal variability in such extreme events has been proposed (Hoerling et al., 2009), such exceptions in explaining vital hydroclimatic extreme events as well as low prediction skill underscore the need to seek explanations beyond current traditional approaches. It is therefore imperative to explore other avenues to improve S2S prediction skill.



Studies have demonstrated that the predictive ability of models may come from their
capability to represent land surface features that have inertia, such as vegetation (evolving cover
and density), soil moisture, snow, among others (e.g., Xue et al., 1996a, 2010b; Lu et al., 2001;
Delire et al., 2004; Koster et al., 2004, 2006; Gastineau et al., 2017). Most land/atmosphere
interaction studies have focused on local effects, for instance, such as those in the previous
Global Land Atmosphere System Study (GLASS) experiment (Koster et al., 2006). The possible
remote (non-local) effects of large-scale spring land surface/subsurface temperature
(LST/SUBT) anomalies in geographical areas upstream of the areas which experience late
spring-summer drought/flood, an underappreciated relation, have largely been ignored.
Observational data in the Tibetan Plateau and the Rocky Mountains have shown that land surface
temperature anomalies can be sustained for entire seasons, and that they are accompanied by
persistent subsurface temperature, snow and albedo anomalies (Liu et al., 2020). Since only 2-m
air temperature (T-2m) has global coverage, and because it is very close to LST in stations with
measurements for both (Liu et al., 2020; also see the discussion in Section 5.1), observed T-2m
data have been used in diagnostic studies to identify spatial and temporal characteristics of land
surface temperature variability and its relationship with other climate variables. Figure 1
exhibits the persistence of the monthly mean difference of T-2m between warm and cold Mays,
which are selected based on a threshold of one-half standard deviation during the period 1981-
2010. Those anomalies can persist for several months, especially during the spring. Preliminary
studies have been carried out to explore the relationship between spring LST/SUBT anomalies
and summer precipitation anomalies in downstream regions in North America and East Asia
(Xue et al., 2002, 2012, 2016b, 2018; Diallo et al., 2019). Data analyses identify significant
correlations between springtime T-2m cold (warm) anomalies in both the Rocky Mountains and
Tibetan Plateau and respective downstream drought (flood) events in late spring/summer.
Modeling studies using the NCEP Global Forecast System (GFS, Xue et al., 2004) and the
regional climate model version of Weather Research and Forecasting (WRF; Skamarock et al.,
2008), both of which were coupled with a land model SSiB (Xue et al., 1991; Zhan et al., 2003)
using observed T-2m and reanalysis data as constraints, have also suggested that there is a
remote effect of land temperature changes in the Rocky Mountains and the Tibetan Plateau on
their respective downstream regions with a magnitude comparable to the more familiar effects of



SST and atmospheric internal variability. Recent studies have further revealed the presence of
LST/SUBT effects in other seasons and regions (Shukla et al., 2019).

The main hypothesis is that LST and SUBT anomalies in early spring carry information about the amount of water locked in frozen ground (i.e., the amount of snow/ice on the ground and in the frozen soil layer below) which is melted in late spring and early summer, as well as the thermal status from the preceding winter which has a long memory. The more snow/ice on the ground and in the frozen soil layer, the longer the seasonal transition from spring to summer. The timing of such a seasonal transition over high elevation areas in the western part (upstream) of the land mass plays an important role in setting up the circulation pattern downstream over the lower elevation areas to the east. The strength as well as the duration of LST/SUBT interactions with downstream circulation patterns should affect the occurrence of droughts or floods in late spring/summer over the eastern part of the continents.

One factor that is closely related to the LST/SUBT anomaly is light absorbing particles (LAPs) in snow. In particular, the snow darkening effect by LAPs in snow due to deposition of aerosols, e.g. desert dust, black carbon and organic carbon from industrial pollution, biomass burning, and nearby wildfires, can reduce snow albedo which increases the absorption of solar radiation by the land surface. This enhanced energy absorption can alter the surface energy balance, leading to anomalous T-2m and snowmelt during the boreal spring. Recent studies have shown that snow darkening effect can lead to large increases in surface temperature over the Tibetan Plateau in April-May, thereby strongly affecting the subsequent evolution of the jet stream and variability of summertime precipitation over India, East Asia and Eurasia (Lau and Kim 2018, Rashimi et al. 2019, Sang et al. 2019). At present, the representation of snow amount, coverage, and LAPs in snow are either absent or grossly inadequate in most climate models, especially in high mountain regions. This could be one of the major reasons for the large diversity in simulated T-2m conditions in current Earth System Models (ESMs).

A number of studies have also started to pursue the potential causes of the spring LST/SUBT anomaly in the Tibetan Plateau and the Rocky Mountains. Analyses based on observational station data over the Tibetan Plateau show that the LST anomaly is highly correlated with anomalous snow, surface albedo and SUBT in the preceding months. Using data from an off-line model incorporating permafrost processes (Li et al., 2010) driven with observed meteorological data as forcing over the Tibetan Plateau, a regression model can predict a LST



anomaly at the monthly and seasonal scales, with surface albedo and middle-layer (40–160 cm) SUBT as predictors (Liu et al., 2020). Additional analyses using observational data show that spring LST in the Tibetan Plateau is significantly coupled with the regional snow cover in preceding months. The latter is also strongly coupled with February atmospheric circulation patterns and wave activity in mid-to-high latitudes (Zhang et al., 2019). Moreover, a modeling

study focusing on North America (Broxton et al., 2017) showed that snow water equivalent (SWE) anomalies more strongly affect April–June temperature forecasts than SST anomalies. It is likely that a temporary filtered response to snow anomalies may be preserved in the LST and SUBT anomalies, and this mechanism deserves further investigation. Additional research on the causes of LST/SUBT anomalies and the association with LAPs in the snow would likely help us

to better understand the sources of S2S predictability.

In the following text, Section 2 introduces the historical development of "Impact of initialized Land Surface temperature and Snowpack on Subseasonal to Seasonal Prediction" (LS4P) and its research objectives. Section 3 presents the LS4P Phase I protocol (LS4P-I): its experimental design and model output requirements. Section 4 discusses causes of current LS4P-

I models' deficiencies in preserving land memory and possible approaches for improvement. Section 5 briefly presents some preliminary LS4P-I results and discusses the future plan and prospectives.

**2. Development of the Initiative on "Impact of initialized Land Surface temperature and**

**Snowpack on Subseasonal to Seasonal Prediction" (LS4P) and its link to other S2S Prediction Programs**

Although T-2m measurement has the longest meteorological observational record with global coverage and the best quality among various land surface variables, its application in S2S prediction has largely been overlooked. Preliminary experiments to test the impact of model

initialization of LST/SUBT on the S2S prediction are encouraging, but the results were obtained from only one ESM and one RCM, with North America and East Asia as the focus regions (Xue et al., 2016b, 2018). Due to the existing shortcomings and uncertainties associated with models, it is imperative to have a multi-model approach in order to further test the LST-memory hypothesis and to explore predictability in more regions. Furthermore, since LS4P proposes a

new approach, involving a decade-long effort to explore, test, and understand the concept, as





well as to develop a proper methodology for the use of ESMs and RCMs, it is also imperative to disseminate information related to the LST/SUBT approach, including lessons-learned and experience, such that more research groups can understand the approach/methodology and test the LST/SUBT effect.

With the preliminary results revealing the promising use of T-2m for LST/SUBT S2S prediction thereby opening a new gateway for improving S2S prediction, the Global Energy and Water Exchanges (GEWEX) and GEWEX/Global Atmospheric System Study (GASS) have supported the establishment of a new Initiative called LS4P. The idea for the new initiative was first presented at the 2nd Pan-GASS meeting in Lorne, Australia, in February 2018. The

initiative was introduced to the GEWEX community at the GEWEX Open Science Conference in Canmore, Canada, May 2018.

        Since the inception of the LS4P in December 2018, more than forty groups worldwide have participated in this effort, including twenty-one (21) ESM groups, many of which are from major climate research centers, nine (9) RCM groups, and seven (7) data groups. A description

of the major components of each of the ESM and RCM models is summarized in Appendix A. A complete listing of LS4P group information can be found at https://ls4p.geog.ucla.edu/. Because LS4P takes a new approach in S2S prediction, GEWEX, the Third Pole Environment (TPE), and the U.S. National Science Foundation have supported two workshops at the American Geophysical Union Fall Meeting in December 2018 and December 2019, and another

one at the Nanjing University, China in July 2019. The workshop goals were to discuss and develop the project, and to provide training for the modeling groups to better understand and practice the LST/SUBT approach (Xue et al., 2019 a, b).

        The LS4P activities are closely related to a number of ongoing international projects. S2S prediction is the topic of a joint project of the World Weather Research Program (WWRP)

& World Climate Research Program (WCRP) which aims to improve understanding and forecast skill at the S2S timescale, between two weeks and a season (WMO, 2013, Vitart et al., 2017; Merryfield et al., 2020). Their S2S project has the study of land initialization and configuration as one of its major activities. The LS4P research activities to address these scientific challenges are consistent with those of the WWRP/WCRP S2S project. The LS4P activity is also closely

related to the TPE program. The TPE has closely worked with LS4P to provide and maintain a data base to support this project. The first phase of LS4P will be a joint effort with the TPE



Earth System Model Inter-comparison Project (TPEMIP), which focuses on regional-scale Earth system modeling over the high elevation Tibetan Plateau region. The LS4P initiative is also relevant to the GLASS Panel because estimating the contribution of land memory to atmospheric

predictability from convective to seasonal timescales is one of its main themes. This requires an understanding of the key physical interactions between the land and the atmosphere, and how feedbacks can change the subsequent evolution of both the atmosphere and the land state. The focus of LS4P on soil temperature also complements GLASS's research on the role of soil moisture as it pertains to land-atmosphere coupling and predictability. LS4P has interacted with

these project groups and developed the experiments which support and complement their planned research activities.

     This LS4P project intends to address the following questions:

•     What is the impact of initializing large scale LST/SUBT and LAPs in snow in climate models on S2S prediction in different regions?

•     What are the relative roles and uncertainties of the associated land processes compared to those of SST in S2S prediction? How do they synergistically enhance S2S predictability?

     LS4P focuses on process understanding and predictability, hence it is different from, and complements, other international projects that focus on the operational S2S prediction. The majority of the models participating in LS4P are ESMs, although, there is a good amount of

RCMs involved. Some difficulties have been identified regarding how to apply RCMs for studying the LST/SUBT effect (Xue et al., 2012). The main concern is that imposition of the same lateral boundary conditions (LBC) for RCM's control and anomaly runs may hamper the necessary modification of circulations at larger scales in the anomaly run. This issue will be more comprehensively studied in LS4P using a much larger RCM domain configuration to

reduce the LBC control on the large-scale change.

     LS4P will organize inter-comparison and validation studies, using satellite, and available ground-based observations, among participating model subgroups in order to pursue a better understanding of the relationship among LAPs in snow in high mountain regions, their deposition, snowmelt/albedo reduction, and the corresponding LST/SUBT anomaly, as well as

how snow darkening affects the S2S predictability.

     The project will ultimately consist of several phases, and each of which will focus on a particular high mountain region on one continent as a focal point. The LS4P-I will investigate



the LST/SUBT effect in Tibetan Plateau. Testing the effect of LAPs in snow is still in the preparation stage and will not be conducted in the Phase I experiments. The second phase of

LS4P will focus on the Rocky Mountains of North America. It is intended that this project will also provide motivation for examining additional high mountains in other continents with similar geographic structure, such as those in South America, for the potential of the LST/SUBT effect to provide added-value to S2S prediction and understanding of the pertinent physical principles.

**3. LS4P First Phase Experiment Protocol: Remote Effects of Tibetan Plateau LST/SUBT**

The Tibetan plateau region provides an ideal geographic location for the LS4P-I test owing to its relatively high elevation and large-scale (areal extent) as well as the presence of persistent LST anomalies. The Tibetan Plateau provides thermal and dynamic forcings which drive the Asian monsoon through a huge, elevated heat source in the middle troposphere, and this has been

reported in the literature for decades (e.g., Ye, 1981; Yanai et al., 1992; Wu et al., 2007; Wang et al., 2008; Yao et al., 2019). A large impact of the Tibetan Plateau LST/SUBT anomaly effect should be expected and has been demonstrated in a preliminary test (Xue et al., 2018).

**3.1 Observational data for LS4P Phase I (LS4P-I)**

There are large amounts of observational data available in the Tibetan Plateau area. The TPE has conducted comprehensive measurements over Tibetan Plateau for more than a decade and has integrated the observational data into the National Tibetan Plateau Data Center (Li et al., 2020), which has more than 2400 different data sets for scientific research focused on the Tibetan Plateau. Featured datasets of high mountainous observations on the Tibetan Plateau

include those from the High-cold Region Observation and Research Network for Land Surface Processes & Environment of China (HORN) which contains the meteorological, hydrological and the ecological datasets (Peng and Zhu, 2017); soil temperature and moisture observations (Su et al., 2011; Yang et al., 2013); multi-scale observations of the Heihe River Basin (Li et al., 2017; Liu et al., 2018; Che et al., 2019; Li et al., 2019); and multiple datasets from the

coordinated Asia-European long-term observing system for the Tibetan Plateau (Ma et al., 2009).

The Third Tibetan Plateau Atmospheric Scientific Experiment (TIPEX-III, Zhao et al., 2018) also provides field measurement data for the LS4P project. The Chinese Meteorological Administration (CMA) provides some field measurements with long term records. The observed



CMA monthly mean precipitation and T-2m, and topography data, with a 0.5-degree resolution
based on station measurements (Han et al., 2019), are used in LS4P to evaluate the LS4P models'
performance over the Tibetan Plateau and to help produce the LST/SUBT mask for model
initialization (see Section 4.2 for details).   There are 80 stations over the Tibetan Plateau
covering the period from 1961-2017. Among them, 14 stations have soil temperature
measurements reaching a depth of 320 cm.  After 2006, more station data are available from the
TPE.  This is in contrast with most ground stations around the world, which only include
measurements for shallow soil layers, e.g., only reaching down to 101.6 cm (Hu and Feng,
2004).  Because of the lack of subsurface measurements, there has been some speculation as to
whether the LST/SUBT anomaly and memory, as well as the hypothesized relationship between
T-2m/LST/SUBT truly exist in the real world.

In addition to the ground measurements, satellite products from 1981 to 2018 from the
Global LAnd Surface Satellite (GLASS, Liang et al., 2013, 2020) data set will also be employed
for this project. This dataset consists of surface skin temperature, albedo, emissivity, surface
radiation components, and vegetation conditions (www.glass.umd.edu).

**3.2 Experimental Design: Baseline and Sensitivity Experiments**
This section describes standard design and configuration for the LS4P-I experiment, which
consists of four tasks.  May and June 2003 are the time periods which have been selected for the
main tests.  The summer of 2003 was characterized by a severe drought over the southern part of
the Yangtze River Basin in eastern China, with an average anomalous precipitation rate of -1.5
mm/day over the area bounded by 112-121°E & 24-30°N[1]. The drought resulted in $100 \times 10^6$ kg
crop yield losses, along with an economic loss of 5.8 billion Chinese Yuan (Zhang & Zhou,
2015). To the north of the Yangtze River, there was above normal precipitation, with
precipitation rates of 1.32 mm/day over the area within 112-121°E & 30-36°N[2]. Over the same
time period, observational data show a cold spring over the Tibetan Plateau; the average T-2m in
May above 4000m was about -1.4°C below the climatological average.  Maximum Covariance
Analysis (MCA, Wallace et al., 1992; Von Storch & Zwiers, 1999) showed a positive/negative
lag correlation between the May T-2m anomaly in the Tibetan Plateau and a June precipitation

---

1        See black box in Figure 6b for reference.
2        See red box in Figure 6b for reference.



anomaly to the south (north) of the Yangtze River. Meanwhile, a preliminary modeling study revealed the causal relationship between the May T-2m/LST/SUBT anomaly over the Tibetan Plateau and the June drought/flood in East Asia (Xue et al., 2018). LS4P intends to further test and confirm such causal relationships with multiple state-of-the-art ESMs in order to assess the uncertainty, and to compare the T-2m/LST/SUBT effect with that of the SST.

(1). Task 1. In Task 1, each modeling group conducts a 2-month simulation starting from around late April to May 1 (e.g., April 27, 28…May 1, …) through June 30, 2003, consisting in a multi-member ensemble. Each group decides whether they use observed May and June 2003 SST and sea ice to specify the ocean surface conditions, which is similar to the AMIP (Atmospheric Model Intercomparison Project) type of experiments, or to use the specific ocean initial condition at the beginning of the model integration (for those ESMs which can run a fully coupled land-atmosphere-ocean configuration), similar to the CMIP (Coupled Model Intercomparison Project) type of experiments, or both. The reanalysis data are used as atmospheric and land initial conditions (as these ESM groups usually do). Since the spin-up time for different models for the S2S simulation varies, some groups start their simulations earlier than May 1, for example, on April 1 or even earlier. LS4P does not require a specific number of ensemble members: each modeling group makes the decision based on their normal practice in performing their S2S simulations, but it is required by LS4P that there should be no less than 6 members. The main purpose of Task 1 is to evaluate the performance of each model for the May 2003 T-2m and the June 2003 precipitation.

The evaluation of Task 1 results will be used to check: (1) model biases in terms of the May 2003 T-2m across the Tibetan Plateau and in terms of June precipitation in the South and North Yangtze River Basins (see the corresponding black/red boxes in Figure 6b as a reference); (2) the lag relationship between these two biases; and (3) the model's ability to produce the observed May 2003 T-2m anomaly in the Tibetan Plateau and the June precipitation anomaly over the areas as listed in criterion (1). The CMA May 2003 T-2m and June 2003 precipitation, these two variables' climatologies, as well as topography data with a 0.5-degree resolution as discussed in Section 3.1 are used to calculate model biases, root-mean-square errors (RMSE), and anomalies; the models' performances are then checked. When calculating the bias, it should be noted that the elevations of the T-2m observational data and model surface are usually not at the same levels, especially in high mountain regions. The observing stations tend to be situated



in valleys and are generally at a lower elevation than the mean elevation of a model grid box.
Before calculating the model bias, the model-simulated T-2m data must be adjusted with a proper lapse rate to the elevation height of the observational data (Xue et al., 1996a; Gao et al., 2017).

The major goals of Task 1 are to check whether most ESMs have large biases in simulating these two variables and whether they are able to produce the observed T-2m and
precipitation anomalies. The relationship between these two biases are evaluated to see whether they are consistent with the observed lag anomaly relationship, i.e., whether a cold/warm bias in May T-2m over the Tibetan Plateau is associated with a dry/wet bias in the South Yangtze River Basin, and an opposite bias to the North of the Yangtze River Basin. The consistency between these relationships would suggest the possibility that reducing the May T-2m bias may reduce
the June precipitation bias. In other words, if a model can produce the observed May T-2m anomaly, it may also be able to produce the observed June precipitation anomaly.

The discoveries from Task 1 will provide crucial information for the LS4P project as it pursues its objectives as discussed in Section 2. If the LS4P ESMs produce no large bias in precipitation and T-2m and/or they are able to simulate the observed anomaly well over Tibetan
Plateau and eastern China, the justification for LS4P would be questionable. Should the model bias relationship between the May T-2m and the June precipitation be the opposite of the observed anomaly relationship of these two variables, it would also be difficult, if not impossible, to pursue the LS4P approach further for these models. The preliminary assessments, however, are encouraging and strongly support the need for LS4P to further pursue its goals, and
they will be briefly demonstrated in Section 5. It should be pointed out that the evaluation of the bias relationship between May T-2m in the Tibetan Plateau and June precipitation in eastern China is just a necessary condition for LS4P to pursue its approach. It is not sufficient to guarantee the model can improve the June precipitation prediction by using improved May T-2m initial conditions. Only Task 3, as discussed below, will serve this purpose.

**(2). Task 2.** A number of LS4P modeling groups are from big climate modeling centers, and, as such, have the required climatological runs already in their respective data bases. Those groups are required to send each year's global May T-2m and June precipitation from their climatological runs. Since different centers have different years in their climatology, LS4P only requires the climatological data set covering the time period from around 1981 to around 2010.



The CMA precipitation and T-2m data averaged over 1981-2010 are employed to assess the simulated climatology biases and RMSE from these groups. The purpose of this task is to check whether the major bias features that we found in Task 1 based on year 2003 for the LS4P ESMs are also present in the modeled climatologies. Our premise is that the large biases in the high elevation Tibetan Plateau region and in the East Asian drought/flood simulation produced by the

LS4P ESMs are also persistent in the models' climatology. As such, any progress achieved in LS4P-I will not be limited to only one individual year, i.e., year 2003, but should have a broader implication. This issue will be further addressed in Section 5.

        **(3). Task 3.** Task 3 is the main LS4P experiment, which tests the effect of the May 2003 T-2m anomaly in the Tibetan Plateau on the June 2003 precipitation anomaly. Thus far, every

ESM has a large bias in producing the observed May T-2m anomaly in the Tibetan Plateau, and so does the reanalysis data, which are used by the ESMs for atmospheric and surface initialization (see more discussion in Section 4.1). To reproduce the observed May T-2m anomaly in the Tibetan Plateau, which is the surface variable interacting with the atmosphere by influencing surface heat and momentum fluxes and affecting upwelling longwave radiation,

initialization of the LST/SUBT has to be improved to generate the T-2m anomaly in the model simulation. Preliminary research suggests that prescribing both LST and SUBT initial anomalies based on the observed T-2m anomaly and model bias is the only way for the current ESMs to produce the observed May T-2m anomalies, unless the observed T-2m is specified during the entire model simulation, which would be a difficult task because, unlike specifying SST, LST

has a large diurnal variation. It should also be pointed out that if we do not impose initial SUBT anomalies in a model simulation, the imposed initial LST anomaly and the corresponding T-2m anomaly would disappear after a couple of days of model integration. Studies based on observational data have shown a high correlation between LST and SUBT, and the memory in the soil subsurface is one of the major factors for producing soil surface temperature anomalies

(Hu and Feng, 2004; Liu et al., 2020).

        To improve the LST/SUBT initialization, a surface temperature mask for each grid point, $\Delta T_{mask}(i,j)$, over the Tibetan Plateau is produced based on each model bias and the observed climate anomaly. The $(i,j)$ represent the latitude and longitude position of the grid point in the model. The initial surface temperature condition at each grid point after applying the mask,

$\mathcal{T}_0(i,j)$, will be defined as follows:



:

$$\tilde{T}_0(i,j) = T_0(i,j) + \Delta T_{mask}(i,j) = T_0(i,j) + \left[n \times T_{obsanomaly}(i,j) - T_{bias}(i,j)\right]$$
$$when\ \overline{T}_{obsanomaly} \times \overline{T}_{bias} < 0 \qquad\qquad (1a)$$

$$\tilde{T}_0(i,j) = T_0(i,j) + \Delta T_{mask}(i,j) = T_0(i,j) + \left[-n \times T_{obsanomaly}(i,j) - T_{bias}(i,j)\right]$$
$$when\ \overline{T}_{obsanomaly} \times \overline{T}_{bias} \geq 0 \qquad\qquad (1b)$$

where $T_0(i,j)$, $T_{bias}(i,j)$, and $T_{obsanomaly}(i,j)$ correspond to the original model surface initial condition, monthly mean model bias, and monthly mean observed anomaly, respectively, at grid
point (i,j). Please note, there are no observed daily data available. The $\overline{T}_{obsanomaly}$ and $\overline{T}_{bias}$ are the averaged observed anomaly and model bias, respectively, over the entire area where the mask is intended to be applied, such as the Tibetan Plateau. Because all current land surface models are unable to maintain the soil temperature anomaly, a tuning parameter "n" (e.g., 1, 2, 3, etc.) is introduced. Through trial and error, each model selects a proper "n" intending to produce the T-
2m anomaly close to observation. For the subsurface, the "n" may be different from that for LST depending on the ESM's land surface scheme. But currently, most modeling groups use the same "n" for every soil layer. This approach can be improved after more deep soil layer measurements are conducted such that we have a better idea about the LST and SUBT relationships. Figure 2 shows schematic diagrams for imposed masks for surface temperature
initialization under different conditions, which shows the concept for the mask formulation. In this figure, a cold year (such as year 2003) and a warm year (such as year 1998) are selected for demonstration. Equation 1, however, can be applied to both cold and warm anomaly years. We use $\overline{T}_{obsanomaly}$ and $\overline{T}_{bias}$ in Equation 1 to determine whether Equation 1a or 1b is used because even if a model has a strong warm/cold bias for the entire area, there are always a few grid points
that have an opposite bias. Using $\overline{T}_{bias}$ in equation 1 as a criterion will prevent the initial conditions of those grid points from adjusting in an opposite direction from the majority of other grid points. Figure 2 should also help serve as a guide to delineate how these grid points' initial conditions would be adjusted based on Equation 1.



Figure 3 shows a mask application example from one LS4P model, which has a warm bias (Figure 3b). Based on the bias and the observed May 2003 T-2m anomaly, a mask using Equation 1 was generated and only imposed over the Tibetan Plateau region (Figure 3c). The mask will be imposed on the initial condition at the first time step of the model integration. The model run will start around May 1 and run through June 30 with multi-ensemble members (the

same total number as for Task 1), and the LST/SUBT will be updated by the ESM after the initial imposition of the mask. However, in the example shown in Figure 3, the mask using n=1 failed to produce proper May T-2m anomaly (Figure 3d). After the model produces a reasonable observed May T-2m anomaly through a tuning of "n" in Equation 1 (in Figure 3, only the mask with n=3 produces proper May T-2m anomaly), the June precipitation difference between the

sensitivity run and the control run from Task 1 will be evaluated.

        To assess the model simulation in this task, we produce composite data sets for global May and June T-2m and precipitation for both the year of 2003 and climatology, in which the CMA data are used for within China for both variables and Climate Anomaly Monitory System (CAMS, Fan and Van den Dool 2008) and Climate Research Unit (CRU, Harris et al., 2014) data

are used for elsewhere for T-2m and precipitation, respectively. These composite data are used to evaluate whether the May T-2m difference between the sensitivity run and the control run produce the observed May T-2m anomaly over the Tibetan Plateau, which is the key objective of Task 3. If a model is capable to produce about 25% of observed May T-2m anomaly over the Tibetan Plateau, we will further examine the difference of the June global precipitation between

the two runs and observed global June precipitation anomaly. Moreover, the improvement in reducing the bias and RMSE for the sensitivity runs will also be assessed.

        **(4). Task 4.** Task 4 tests the effect of the SST on the June 2003 precipitation. There are two possible approaches for this test. Groups with the AMIP type of experiment use the observed May and June 2003 SST for their Task 1 and Task 3 experiments. For those groups, in

Task 4, the 2003 SST conditions will be replaced by the climatological SST. For modeling groups with the CMIP type experiment, the 2003 initial condition used in Task 1 and Task 3 will be replaced by the climatological initial condition. The year 2003 is a La Niña year. The modeling groups with the CMIP type of simulations need to check their models' SST simulations to be sure that their models are producing adequate La Niña conditions along the western coast

of South America and the eastern Pacific. The June precipitation difference between control run





(with 2003 SST) and the Task 4 run (with climatology SST) will be compared with the observed anomaly in 2003 to assess the global SST effect on the precipitation and compared it with the LST/SUBT effect from the Task 3 results.

**(5). Model Output and Availability**

The data output requirements take into account the evaluations that are required as discussed in Sections 3.2 (1)-(4), as well as information required to characterize the land surface/atmosphere interactions at and near the surface, and the mid and upper troposphere atmospheric wave propagation. In addition to the T-2m and precipitation, other model output

from the land surface and the atmosphere will also be used to evaluate the model results. The NOAA metrics and protocol for short to medium range weather forecast performance evaluations as discussed in Wang et al. (2010) will be applied to assess model performance. Careful considerations are necessary to limit output frequency in order to save storage while still providing sufficient information for crucial diagnostic analyses. The LS4P data are stored and

will be distributed through the National Tibetan Plateau Data Center (Li et al., 2020) and the U.S. Department of Energy Lawrence Livermore National Laboratory Earth System Grid Federation (ESGF) node (Cinquini et al., 2014). The detailed information is described in Appendix B.

**4. Main Issues in LST/SUBT initialization**

To date, all of the LS4P ESMs with their land models have difficulty producing the observed T-2m anomaly over the Tibetan Plateau to varying degrees. In other words, they are unable to maintain the imposed LST/SUBT anomaly from the mask during the model integration. The current model deficiencies in T-2m simulation are rooted in the data, mainly from the reanalysis data, which are used for

the model initialization, and the model parameterizations, mainly in the land module.

**4.1 Data Uncertainty**

Observational T-2m/LST/SUBT data are crucial for model initialization of surface conditions and for model validation. However, ground measurements over high-elevation areas are relatively sparse. For

instance, most currently available gridded global T-2m data sets with long records only consist of a few dozen stations over the Tibetan Plateau. Considering the complex topography of the region, potentially



large interpolation errors can occur. The same is true for the reanalysis data, which are used for the model initialization. In most reanalysis data sets, the T-2m is only a model product. In LS4P, we employ the CMA T-2m data (1980-2017) with 0.5-degree resolution (Han et al., 2019) to use for model initialization,

and it is based on about 150 ground station measurements over the Tibetan Plateau. Figure 4 shows the May T-2m climatology (the 1980-2013 average) over the Tibetan Plateau, and the anomalies of May 2003/1998, which corresponds to a very cold/warm spring in the Tibetan Plateau, respectively, from CMA, CAMS, CRU, Climate Forecast System Reanalysis (CFSR, Saha et al., 2014), ERA-Interim (ERAI, Berrisford et al., 2011), and the Modern-Era Retrospective analysis for Research and Applications,

version 2 (MERRA-2, Gelaro et al., 2017). Because each T-2m data set has its own elevation, all of the data have been adjusted to the CMA elevation for comparison. Compared with the CMA data, the CAMS/CRU climatology is about 1.8°C cooler/1.5°C warmer, respectively. The biases for warm/cold years would be even larger for CAMS/CRU (not shown), respectively. While the climatological bias for CFSR reanalysis data is small, the bias for ERAI is still on the order of one standard deviation of the

Tibetan Plateau T-2m variability (~0.7 °C). The bias is larger in MERRA-2, at about 4°C. In addition, for cold/warm years, MERRA-2 and CFSR show opposite anomalies. The large surface temperature biases in the reanalysis data sets likely interact with temperature of the lower atmosphere. There are limited atmospheric sounding data over the Tibetan Plateau for data assimilation. That said, lower atmosphere temperature is also subject to model bias. Since there are no observed near surface layer observations, we

compare the reanalysis surface and near surface temperature anomalies with their own climatology. These anomalies are very close (not shown), which means even if we impose a mask to overcome the LST/SUBT bias, the bias in the lower troposphere is still there. This bias in the reanalysis data has an important implication in affecting the LST initialization and its simulation, which will be discussed further in section 4.2.

In addition to the surface temperature, subsurface temperature initialization is also challenging in high elevation areas. Measurements for deep subsurface conditions do not exist in most mountain areas. However, there are fourteen stations in the Tibetan Plateau (Figure 5a) that have soil temperature measurements during the period 1981-2005 at depths of 0, 5, 10, 15, 20, 40, 80, 160, and 320 cm, which shed light on the quality of subsurface layer temperature in the reanalysis data. Below 320 cm, the soil

temperature exhibits very little annual variation. The soil temperature profiles from station observations are averaged and four typical months that represent the four seasons are displayed in Figure 5b. The differences between the T-2m and the LST are less than 1 degree for these four months. During winter and





summer, the deep soil temperature profiles show a larger lag compared with the LST. The reanalysis products over the grid points closest to the observation stations (Figure 5a) have been averaged over the

same time period. However, these data show large discrepancies compared to observations in addition to biases (Figures 5b-c). For instance, the top 1-m soil temperatures in the ERAI data are nearly constant for every season with little change with soil depth. In MERRA-2, the lag response in the soil profiles only appears in the winter and summer up to about 1 m deep; for other seasons or soil temperature below 1-m does not change much. The CFSR shows a better lag response, but it only reaches 1.5 m in depth. Its

biases in these stations compared to the observation stations are also apparent.

The deficiencies in the reanalysis products pose a challenge for properly producing the observed T-2m anomalies since the reanalyses are used to provide the basis for the surface initial condition for most ESMs. Since every LS4P ESM showed a large bias in simulating the May 2003 T-2m anomaly over the Tibetan Plateau, we have addressed how to take the bias into account in producing the initial condition

mask in Section 3.2. In the next section, the efforts from different modeling groups to generate the observed T-2m anomaly will be presented further.

### 4.2 Approaches in Improving the LST/SUBT Initialization and T-2m Anomaly Simulation

In addition to the data that are used for LST/SUBT initial conditions, land models also have deficiencies in

maintaining the anomalies that are imposed using an initial mask as discussed in Section 3.2. In the LS4P-I experiment, most models are only able to partially produce the observed T-2m anomaly in May despite the fact that the imposed initial masks normally contain much larger anomalies than those observed. This section highlights some specific approaches undertaken by a few groups during their application of the LS4P-I protocol to improve the T-2m anomaly simulation.

The surface soil (20-30 cm) in the central and eastern Tibetan Plateau contains a large amount of organic matter which greatly reduces the soil thermal conductivity and increases the soil heat capacity (Chen et al., 2012; Liu et al., 2020). However, this factor is not taken into account in the LS4P ESMs, except CNRM-CM6-1. That said, the soil thermal conductivity/heat capacity over the Tibetan Plateau in the ESMs is too high/too low. In addition, some ESMs overestimate the precipitation over the Tibetan

Plateau, making the soil water content higher than in reality (Su et al., 2013), which also leads to higher soil thermal conductivity. Less soil organic matter and high soil moisture both accelerate the heat exchange rate between the soil and the atmosphere, which causes the rapid loss of soil thermal anomalies in the models.





The soil layer depth in the ESM also affects the model's ability to generate the observed T-2m

anomaly. The long memory in deeper soil helps to preserve the soil temperature anomaly in shallower layers. In a sensitivity study that changed the soil depth from 6 m to 3 m, it was found that with reduced total soil column depth, a similar magnitude anomalous soil temperature can only be kept for about 20 days, then it disappears much faster thereafter compared with the 6-m soil layer model (Liu et al., 2020). In a number of LS4P land models, the total soil column depth is less than 3 m. To overcome these

shortcomings in current ESMs and to reproduce the observed T-2m anomaly, a tuning parameter "n" is introduced (Eq. 1) when setting up the surface mask since it is not a simple task to increase the soil layer depth for all of the ESMs.

One of the intentions of the initialization of LST/SUBT is to influence the lower atmosphere since the corresponding initial condition from reanalysis also has inherent errors as discussed in section 5.1, and

for some models they can be quite large. A number of modeling groups have started the model simulation earlier, for instance on April 01, in order to have sufficient time for the lower atmosphere to spin-up and to be consistent with the within-mask imposed soil surface conditions. In some models, such as ACCESS-S2 and KIM, the models make an adjustment after reading in the initial condition, usually referred to as shock adjustment, in order to avoid an imbalance between the atmosphere, land, and ocean initial conditions.

This shock adjustment has become a more popular practice in a number of modeling groups. The idea behind the shock adjustment arises from the potential inconsistency among different sources for initial conditions, and the belief that the atmospheric components are considered to be relatively the most reliable. With such an approach, within the first week or 10 days, the atmospheric forcing plays a dominant role in adjusting the other components' initial conditions. As such, the imposed initial soil temperature from the

mask at the top soil layers could be compromised very dramatically toward the lower atmospheric conditions, which, unfortunately, also have large errors over theTibetan Plateau as previously discussed. Although the imposed deep soil temperatures eventually start to affect the air temperature, this process generally takes more than 20 days. For the model with such a shock adjustment, the mask needs to be imposed when the shock adjustment becomes weak, such as at the second day in ACCESS-S2 or half a

month after the initial simulation date, as done in KIM. As such, the models may have to start their integrations much earlier. A couple of models tried to impose the mask more than once to produce the T-2m anomaly. For instance, the FGOALS-f2 model imposed the LST/SUBT anomaly on both May 1 and May 2 to better produce the observed T-2m anomalies. It should be pointed out that if a mask is imposed too many times, the ΔT in the mask may add up every time when it is imposed to become quite large



sink/heat source. Furthermore, enforcing the LST/SUBT perturbation too many times during the model simulation with accumulated large ΔT may distort the atmospheric conditions. Precautions must be taken in this type of approach, probably with ΔT imposed no more than twice with a well-designed scheme to avoid the excessive accumulation of heating/cooling.

For the E3SM and CESM2, which are mainly used in long-term climate research (e.g., century-
long simulations), real time initialization for S2S prediction is not very closely related to the research objective the model centers intend to pursue. To conduct LS4P type research, the modeling groups have to develop an approach in nudging the reanalysis data for a real time initialization. Nudging is one of the simplest data assimilation methods (Hoke and Anthes, 1976) and has been widely used in climate model evaluation and sensitivity studies (e.g., Xie et al., 2008; Sun et al., 2019; Tang et al., 2019) to constrain the
simulations towards a predefined reference (the reanalysis data in this case) and hence to facilitate time-specific comparisons between model and observations. For the LS4P simulations, E3SM and CESM2 used 1-month worth of nudging of the horizontal wind components (U & V) with a 6-hour relaxation time scale before the land mask for the initial LST perturbation was applied. A study (Ma et al., 2015) has shown that nudging only horizontal winds produces better results compared with those with nudging of
more variables, such as temperature, specific humidity, etc.

## 5. Discussion: Prospective and Impact of LS4P

LS4P is the first international grass-root effort focused on introducing spring LST/SUBT anomalies over high mountain areas as a factor to improve S2S precipitation prediction through the remote effects of
land/atmosphere interactions. Although the original idea of starting LS4P was more limited and only aimed at evaluating whether the results from preliminary tests with one ESM and one RCM (Xue et al., 2016b, 2018) could be reproduced by more modeling groups, multi-model participation has quickly lead to the recognition that the Tibetan Plateau's spring LST/SUBT effect on the precipitation anomaly to the south and north of the Yangtze River was only a small part of broader aspects.

Figure 6 shows the observed May T-2m and June precipitation anomalies in 2003 and the corresponding ensemble mean biases from 13 LS4P ESMs for these two variables in 2003 over the eastern part of Asia. Among the 13 ESMs, eleven ESMs had warm T-2m biases while the remaining two had cold biases, respectively. Because the May 2003 T-2m had a cold anomaly, the T-2m and precipitation biases for the models with positive T-2m bias were multiplied by -1 to produce the ensemble mean composites as
shown in Figures 6c and d. Despite very different data sources (observed T-2m data were from CMA over





China and CAMS for regions outside of China, observed precipitation data were from CMA over China and CRU in regions outside of China), and the fact that ESMs results were produced from models with different numerical approaches and physical parameterizations, the modeled bias relationships between May T-2m and June precipitation are very consistent with the observed anomaly relationship between

observed May 2003 T-2m over Tibetan Plateau and June 2003 precipitation in many parts of eastern Asia, in addition to the Yangtze River basin. For instance, models with a cold bias in May T-2m in the Tibetan Plateau also have a dry bias in June precipitation over Northeast Asia, part of southeast and South Asia, and Siberia, and a wet bias to the west of Siberia, consistent with the observed precipitation anomaly. The models with the opposite sign of T-2m bias produced the opposite precipitation response. The correlations

between observed June precipitation anomalies and the corresponding model biases are 0.62. Furthermore, the T-2m cold bias over the Tibetan Plateau is associated with a cold bias in the Iranian Highlands and a warm-cold-warm wave train over the Eurasian continent, which is also generally consistent with the observed T-2m anomalies. Moreover, the consistencies suggest a possibly much larger scale remote effect of the Tibetan Plateau LST/SUBT on summer precipitation over many parts of the

world and support the LS4P's approach in its experimental design as discussed in Section 3.2. As a result, the diagnostic analyses from the tasks in Experiment 1 will cover the entire globe. Comprehensive analyses and discussion will be presented in subsequent papers after the LS4P groups have completed their experiments.

Although the T-2m anomaly covers large areas, our previous study has shown that only the

LST/SUBT anomaly over high mountains had a substantial impact on the subsequent drought (Xue et al., 2012). One of the LS4P groups, KIM, also tested the effect of the LST anomaly in other parts of East Asia, but found their effects are incompatible with the Tibetan Plateau LST/SUBT effect. In addition to year 2003, we also checked the May T-2m and June precipitation bias in the climatologies of the different models. The thirteen ESMs shown in Figure 6 have also provided their climatological data sets. Figure 7

shows the climatological biases for May T-2m and June precipitation from these ESMs. The patterns between the bias in the 2003 simulation and the bias in the model climatologies are generally consistent, which suggests that the findings from the 2003 case may have a broader implication.

In Phase I, through the LS4P RCM efforts in incorporating the TPE and TIPEX-III data, we also intend to adequately simulate water and energy cycle and atmospheric conditions in the Tibetan Plateau

and their variability. These simulations will provide the data for better atmospheric and surface





initialization, along with obtaining an improved understanding of the atmospheric circulation and water cycle in Tibetan Water Tower.

Thus far, our discussion has been focused on the modeling approach. A recent statistical study has shown that spring soil temperature in central Asia could be a predictor of summer heat waves over northwestern China (Yang et al., 2019). In addition, surface temperatures from five Northern European observing stations have been used as a predictor for long-range forecasting of southwest monsoon rainfall over India (Rajeevan, et al., 2007). Moreover, spring (April-May) precipitation and 2m air temperature over northwestern India, Pakistan, Afghanistan, and Iran have been found to have a strong link with the first phase (June-July) of summer monsoon rainfall over India (Rai et al., 2015). We will extend the data analyses for different major mountains and different seasons and to identify hot spots over the globe where LST has significant impacts. Preliminary statistical forecasts will also be explored, using methods such as the Canonical-Correlation Analysis (CCA) and Joint Empirical Orthogonal Analysis (JEOF) (Smith et al., 2016). Based on the statistical analyses, a Tibetan Plateau Oscillation Index (TPO) and a Rocky Mountain Oscillation Index (RMO) will be proposed for predictions of the hydroclimatic extreme events, and a relationship between the TPO and the RMO indexes will also be investigated. As discussed in Section 3, the Rocky Mountain LST/SUBT effect will be the focus of LS4P Phase II (LS4P-II).

The LS4P research has revealed some severe deficiencies in current land models in preserving the land memory. In many models, the force-restore method (Deardorff, 1978; Dickinson, 1988; Xue et al., 1996b) is used to represent subsurface heat transfer and soil thermal status. This simple method produces adequate diurnal and seasonal cycles of surface temperature and thus has been widely used by many land models for decades. However, its severe deficiency in keeping the soil memory is apparent in the LS4P studies. We have found that excessively shallow soil depths along with simplified parameterizations of subsurface heat transfer are acting to limit the soil memory effect in many models, especially in cold regions. An innovative approach has been developed for the land model initialization that can help maintain the monthly LST/SUBT anomaly. The LS4P's finding on why ESMs have difficulty to maintain the LST anomaly, and its proposed approach to help solving the issue should be a significant contribution from the LS4P project to improve the S2S prediction.

One issue that hampers the application of the LST/SUBT approach for S2S prediction is data availability. The TPE has conducted comprehensive measurements over the high mountain Tibetan Plateau areas, which include a plateau-scale observation network plus intensive networks at



more local scales: these data consist in boundary-layer observations, land surface and deep soil layer measurements. These measurements have provided invaluable information to support the establishment of the LS4P and to foster further development. Currently, such comprehensive

measurements over high mountain areas are still lacking across the globe. GEWEX has been planning for more measurements that are related to land/atmosphere interactions (Wulfmeyer et al., 2020; Schneider and van Oevelen, 2020). We hope that the results from LS4P will demonstrate the substantial role of high mountain surface conditions on global climate and atmospheric circulation, and therefore stimulate more initiatives to increase land/atmosphere interaction measurements over high

mountain regions.

LS4P will complete the Phase I tasks at the end of 2020. A special issue in Climate Dynamics has been initiated in late 2020 to report various LS4P research results and other S2S prediction research results that should help increase the understanding and predictions of land-induced forcing and atmosphere interactions on droughts/floods and heatwaves. A possible LS4P article in a high impact journal will also

be prepared. We plan to kick-off the LS4P-II in the summer or later of 2021 with a workshop at the Earth System Science Interdisciplinary Center (ESSIC), University of Maryland, College Park, USA. This workshop will summarize the phase I activity and design working tasks for the LS4P-II.

Although land has lower heat capacity and less moisture compared to the oceans, the land surface has a much stronger response to changes in surface net radiation at diurnal, sub-seasonal,

and seasonal scales compared to oceans. This is particularly true in high elevation areas, which could provide a useful source for predictability at these scales. LS4P intends to improve the S2S precipitation prediction through a better representation of land surface processes in the current generation of ESMs and aims to make a fundamental contribution in advancing S2S prediction through proper initialization of LST/SUBT in high mountain regions. The LS4P approach proposes a new

front in S2S prediction to complement other existing approaches. We hope activities and results from LS4P-I can provide a prototype approach to raise further scientific questions and open a new gateway for more studies with various approaches to better understand the roles of different forcing and internal dynamics in S2S predictability along with identifying the relevant mechanisms.



# Appendix A: List of LS4P-I Earth System Models (ESMs) and Regional Climate Models (RCMs)


## Table A1: List of LS4P-I Earth System Models

| Model | Institution Name | Contact personnel | Resolution | Convection scheme | PBL | Land Surface | Aerosols/dust |
|---|---|---|---|---|---|---|---|
| **ACCESS-s1/s2** (MacLachlan et al. 2015) | Bureau of Meteorology, Australia | Maggie Zhao | N216L85, ocean 0.25 | Mass Flux | Adrian lock | JULES | None |
| **AFES ver 4.1** (Nakamura et al. 2015) | Hokkaido University, Japan | Tetsu Nakamura | T79 (~150km), and 56 vertical level up to about 0.1hPa | Emanuel convection scheme | Nonlocal boundary layer scheme | MATSIRO | Sekiguchi (2004) |
| **BCC-CSM** (Wu et al. 2019) | National Climate Center, China Meteorological Administration, China | Xueli Shi Weiping Li | T106 (Atmosphere: 110km Ocean: 30km) | Hack (1994), with modified deep convection scheme (Wu et al. 2019) | Holtslag and Boville (1993) | BCC-AVIM2.0 | Prescribed |
| **BESM** (Nobre et al. 2013) | National Institute for Space Research (INPE), Brazil | Paulo Nobre | Atmos: T062L42 Ocean: 1deg lon varying lat: 1/4 equator ½ poles | Arakawa | Bretherton and Park (2009) | IBIS/SIB | Climatological Horizontally varying |
| **BNU-ESM** (Ji et al. 2014) | Beijing Normal University (BNU), China | Tianyi Fan Duoying Ji | 1.9° x 2.5° | modified Zhang–McFarlane scheme | non-local diffusion | Common Land Model (CoLM; Oleson et al. 2010) | None |
| **CAS-ESM** (Lin et al. 2016) | Institute of Atmospheric Physics, Chinese Academy of Sciences, China | Lin Zhaohui Yanling Zhan | 1.4°×1.4° | Modified Zhang-McFarlane | UW diagnostic TKE | CLM4.0 (Oleson et al. 2013) | Modal Aerosol Model |
| **CAS-FGOALS-f2** (Bao et al. 2019) | BNU and IAP/LASG, China | Xin Qi Jing Yang Qing Bao | 100km | Resolve Convective Precipitation (RCP) | the University of Washington moist turbulence (UWMT) scheme | CLM4 (Oleson et al. 2013) | prescribed |
| **CESM2** (Danabasoglu et al. 2020) | The University of Arizona, | Michael Brunke | ~0.9°x1.25° | Deep (Zhang and McFarlane 1995) | CLUBB | CLM5 (Lawrence et al. | MAM4 (Liu et al. 2016) |



| | | | | | | | |
|---|---|---|---|---|---|---|---|
| | USA | | | Shallow by CLUBB (Golaz et al 2002) | | 2019) | |
| **CFS/SSiB2** (Xue et al. 2004; Lee et al. 2019) | University of California – Los Angeles, USA | Ismaila Diallo Yongkang Xue | T126 (~1ºx1º) & 47 vertical levels | Simplified Arakawa–Schubert (SAS) | Nonlocal boundary layer scheme | SSiB2 (Xue et al. 1991) | Prescribed fixed |
| **CIESM** (Lin et al. 2019,2020) | Tsinghua University, China | Yi Qin Yanluan Lin | 1º x1º & 30 vertical levels | Modified Zhang–McFarlane | Bretherton and Park (2009) | CLM4.0 (Oleson et al. 2013) | Prescribed following MACv2-SP |
| **CMCC-SPS3** (Sanna et al. 2016) | Fondazione Centro euro-Mediterraneo sui Cambiamenti Climatici (CMCC), Italy | Stefano Materia Daniele Peano | ne30np4 (~ 111km grid spacing at the equator) & 46 atmospheric vertical levels up to 0.2 hPa | Park and Breterthon (2009) | Bretherton and Park (2009) | CLM4.5 (Oleson et al. 2013) | Aerosol prescribed to a 2000 climatology; SNICAR |
| **CNRM-CM6-1** (Voldoire et al. 2019) | CNRM, France | Constantin Ardilouze, Aaron A. Boone | Tl127 (~ 150 km), and 91 levels up to 0.01 hPa | PCMT | Turbulence: Cuxart et al. (2000) | ISBA-CTRIP | prescribed to a climatology |
| **ECMWF-IFS** Version: CY46R1 (Johnson et al. 2019) | ECMWF, United Kingdom | Retish Senan, Frederic Vitart, Gianpaolo Balsamo, Patricia de Rosnay | Atmos: Tco199 (~25 km), 91 vertical Levels Ocean: ORCA025 (~25 km) 75 vertical levels | Based on original Tietdke scheme with several improvements | McRad radiation scheme | HTESSEL scheme | CMIP5 forcing |
| **E3SM** (Golaz et al. 2019) | Lawrence Livermore National Laboratory, USA | Qi Tang Shaocheng Xie Yun Qian | 1º x 1º for atmosphere and land | Shallow Conv.: CLUBB Deep Conv.: ZM | CLUBB | ELMv0 Note: this is our land model. | Flanner et al. (2013) |
| **GEFSv12** (Zhou et al. 2019) | EMC/NCEP/ NOAA, USA | Yuejian Zhu, Hong Guan, Wei Li | 0.25º (~25 km) | updated scale-aware SAS convective parameterizations (Han et al. 2017) | K-EDMF PBL scheme | Noah Land Surface model (Ek et al. 2003) | Inline aerosol representation based on GOCART |
| **GEM-NEMO** (Smith et al. 2018) | Environment and Climate Change | Hai Lin Ryan Muncaster | 1.4º x 1.4º; 79 vertical levels (L79) | Kain-Fritch scheme for deep convection, Kuo-transient | 1.5 order closure E-L | ISBA | None |



| | | | | | | |
|---|---|---|---|---|---|---|
| | Canada, Canada | | | scheme for shallow convection | | | |
| **GRAPES_GFS** (Chen et al. 2020) | China Meteorological Administration, China | Zhang Hongliang | 0.5º | NSAS | NMRF | COLM | Climate data |
| **IITM CFS** (Saha et al. 2014, 2017) | Indian Institute of Tropical Meteorology, Ministry of Earth Sciences, India | Subodh Kumar Saha | T126 (~1ºx1º) &47 levels, up to 0.01 hPa | Simplified Arakawa–Schubert (SAS) | Nonlocal boundary layer scheme | NOAH (Ek et al. 2003) | Prescribed fixed |
| **JMA/MRI-CPS-2** (Takaya et al. 2018) | Japan Meteorological Agency/Meteorological Research Institute, Japan | Yuhei Takaya | Atmosphere: 110km, Ocean: 1° x 0.3-0.5° | Arakawa-Schubert scheme | Mellor-Yamada Level 2, Monin-Obukov similarity | Simple Biosphere model (JMA-SiB) | Climatology |
| **KIM** (Song-You et al. 2018; Hong et al. 2018) | Korea Institute of Atmospheric Prediction Systems, South Korea | Myung-Seo Koo Song-You Hong | T126L42 (~111km) | KIM SAS (KSAS; Han et al 2020) | Scale-aware YSU (Lee et al. 2018) | Revised NOAH LSM (Koo et al. 2017) | Prescribed climatology (Choi et al. 2019) |
| **NASA_GEOS5** (Molod et al. 2020) | NASA Goddard Space Flight Center, USA | Hailan Wang | 1-degree | Relaxed Arakawa-Schubert scheme | Lock scheme combined with Louis and Geleyn algorithm | Catchment land model | GOCART aerosol model that predicts dust, sea salt, sulfate, nitrate, organic carbon, and black carbon |



**Table A2: List of LS4P-I Regional Climate Models**

| Model | Institution Name | Contact personnel | Resolution | Conv. scheme | PBL | Land Surface | Aerosols/dust |
|---|---|---|---|---|---|---|---|
| **CWRF** (Liang et al. 2012) | University of Maryland, College Park, MD, USA | Xin-Zhong Liang Haoran Xu | 30 km | Ensemble Cumulus Parameterization (ECP) penetrative convection (Qiao and Liang 2016) plus UW shallow convection (Bretherton and Park 2009) | CAM (improved Holstag and Boville 1993) | Conjunctive Surface-Subsurface Process (CSSP) | Prescribed MODIS aerosol data |
| **Eta RCM** (Mesinger et al. 2012) | National Institute for Space Research (INPE), Brazil | Sin Chan Chou Jorge Luis Gomes | 40 km & 38 vertical layers | Betts-Miller-Janjic (Betts and Miller 1986; Janjic 1994) | Janjic 2001 | NOAH (Ek et al., 2003) | Constant effect/no dust |
| **RegCM4.3-CLM4.5** (Wang et al. 2016) | University of Connecticut (UCONN), USA | Guiling Wang Miao Yu | 50 km & 23 vertical layers | MIT-Emanual (Emanuel 1991) | Holstag (Holstag et al., 1990) | CLM4.5 (Oleson et al., 2013) | None |
| **RegCM4.6.1** (Giorgi et al. 2012) | Nanjing University, China | Jianping Tang Shuyu Wu Weidong Guo | 20 km | Tiedtke (Tiedtke, 1989) | Holstag (Holstag et al., 1990) | CLM3.5 (Oleson et al. 2013) | None |
| **WRF-CHEM** (Grell et al. 2005) | Sun Yat-sen University, China | Zhenming Ji | 25 km | Grell-Devenyi (Grell and Dévényi 2002) | Mellor-Yamada-Janjic (Schaefer, 1990) | Noah (Ek et al 2003) | CBMZ (Zaveri and Peters, 1999); MOSAIC (Zaveri et al., 2008) |
| **WRF V3.8.1** (Skamarock et al. 2008) | Institute of Atmospheric Physics, Chinese Academy of Sciences (IAPCAS), China | Yuan Qiu Jinming Feng | 25km | New Simplified Arakawa-Shubert (Han et al., 2020) | Yonsei University Scheme (Hu et al., 2013) | SSiB (Xue et al., 1991) | None |
| **WRF v3.9** (Skamarock et al. 2008) | Institute of Tibetan Plateau-Chinese Academy of Science (ITP- | Xu Zhou KunYang | Domain01 : 0.24 degree Domain02 : 0.08 degree | no | Mellor–Yamada–Janjic turbulent kinetic energy (TKE) | Noah (Ek et al. 2003) | None |



| | CAS), China | | | | | | |
|---|---|---|---|---|---|---|---|
| **WRF v3.9.1.1** (Skamarock et al. 2008) | Japan Agency for Marine-Earth Science and Technology (JAMSTEC), Japan | Shiori Sugimoto Tomonori Sato Hiroshi Takahashi | 20km | Grell 3D ensemble scheme | MYNN 2.5 level TKE scheme | Unified Noah land-surface model (Ek et al. 2003) | None |
| **WRF v4.1.3** (Skamarock et al. 2008) | Department of Atmospheric Sciences, Yonsei University, South Korea | Jinkyu Hong Jeongwon Kim | 15 km and 61 vertical layers to 50 hPa | Grell-Freitas Ensemble scheme | Yonsei University (YSU) scheme + canopy height + Roughness sub-layer scheme (Lee and Hong 2016) | Noah (Ek et al. 2003) | None |




## Appendix B. Model Output and Availability

Five types of variables are requested: they include monthly and daily mean 3-dimensional atmospheric profile variables at 1000, 925, 850, 700, 600, 500, 300, 200, and 100 hPa, as well as
monthly, daily, and 6-hourly/3-hourly 2-dimensional surface variables. The detailed variable requirements are listed in the Supplemental Table S1. Since LS4P-I explores the timescales necessary for realistic simulation of sub-seasonal and seasonal (S2S) weather and climate phenomena, a minimum amount of sub-daily data is required to allow the diagnosis of phenomena related to S2S and monsoon systems. These model outputs are generally consistent
with the requirements of the NOAA metrics and protocol for short to medium range weather forecast performance evaluations. If a model does not output one of the requested variables, it should report it as a missing value. Due to the nature of the LS4P project, daily surface temperature and precipitation data must be included, especially surface temperature data which will be used to check and improve the model performance with respect to its ability to reproduce
the observed T-2m anomaly. Finally, only ensemble means are required.

The LS4P data are stored and will be distributed through the National Tibetan Plateau Data Center (http://data.tpdc.ac.cn/en/) and the U.S. Department of Energy Lawrence Livermore National Laboratory Earth System Grid Federation (ESGF) node (https://esgf-node.llnl.gov/projects/esgf-llnl). The National Tibetan Plateau Data Center has an online data
submission system similar to that used for paper submission. For instance, folders can be uploaded without being tarred into a single file. It is also recommended that each modeling group create its own folder, which may contain many subfolders/files, using labels such as Task1, Task2, etc., under which it is suggested to create more subfolders for the monthly, daily, and 6-hourly data, respectively.

Data files must comply with the NetCDF format, version 4. The names of the files in the LS4P archives should follow the example below and must appear in the following order: VariableName_LS4P_ESMModelName_LS4PExperimentName_Frequency_[StartTime-EndTime].nc. For example, the file name, pr_LS4P_UCLACFSSSiB2_Task1_6hr_00z01052003 -18z30062003.nc, represents the precipitation data from Task1 using the
UCLA CFS/SSiB2 model and covers the period from 01 May 2003 through 30 June 2003 (i.e. the date is recorded as ddmmyyyy). A document that specifies the technical aspects of



LS4P data archive and data formats, including the common naming system, is provided in Appendix C.



**Appendix C: LS4P-I Data Archive Design**


This appendix specifies technical aspects of LS4P-I data archive and data formats, including the common naming system. The List of requested LS4P-I variables and time-scale is contained in "LS4P_ESM_outputs_list_update" available from

https://ls4p.geog.ucla.edu/experiments/. But it could also directly be downloaded from the following link: https://ucla.box.com/s/oeo8yq9jx58im4mlfd5lgbnl42ewk180.

**I)        File Format and File Naming**

Only ensemble means are required to submit to the data base. Data files have to comply with the NetCDF format, version 4. The names of the files in the LS4P-I archives are made as

describe below and must appear in the following order:

**VariableName_LS4P_ESMModelName_LS4PExperimentName_Frequency_[StartTime-EndTime].nc**

**VariableName corresponds to the name of the target variable in the NetCDF files.**

**ESMModelName** identifies the model name.

**LS4PExperimentName** identifies the experiment name [Task1], [Task2], [Task3] and [Task4]. Task3 is for LST/SUBT experiment. If you use different CTRL for Task3 other than Task1, please use [Task3-CTRL] to identify theTask3 control run. In case you need clarification about, please contact us.

**Frequency** is the output frequency indicator: 3hr=3hourly, 6hr=6hourly, day=daily,

mon=monthly.

**StartTime** and **EndTime** indicate the time span of the content of the file, such as 00z01052003 and 18z30062003. For example, pr_LS4P_UCLACFSSiB2_Task1_3hr_00z01052003-18z30062003.nc

**II)        Uploading/Acquiring LS4P-I Data Procedure in the National Tibetan Plateau**

**Data Center**

The data portal is available at http://data.tpdc.ac.cn/en/. The login is "LS4P_group". National Tibetan Plateau Data Center has an online data submission system which is similar to paper submission system. For instance, folders can be uploaded, but not needed to be tarred in one file. It is recommended that each modeling group to create its own folder using

the following naming: **InstituteName_ESMModelName** (example: UCLA_CFS-SSiB2).





Note that each folder can contain many subfolders/files (e.g. UCLA_CFS-SSiB2/Task1/ or UCLA_CFS-SSiB2/Task2/…). It is recommended to create subfolder for each **LS4PExperimentName (example: Task1, Task2, Task3, and Task4).** Additionally, under each LS4PExperimentName subfolder, we suggest creating subfolders such as monthly,

daily, and 6hourly (e.g. UCLA_CFS-SSiB2/Task1/monthly/ or UCLA_CFS-SSiB2/Task1/daily/…).

**A)** **Uploading Data into National Tibetan Plateau Data Center using Filezilla**

To upload data into the National Tibetan Plateau Data Center, we recommend to use "Filezilla". With Filezilla, the host, username and password are generated automatically for

the Filezilla when the data are uploaded. The following procedure is based on "Filezilla".

The procedure will utilize the following steps.

**1).** Log into the online National Tibetan Plateau Data Center (http://data.tpdc.ac.cn/en/), using the aforementioned login details (see II). Login name: LS4P_group.

**2).** Go to 'LS4P_group'/'personal center'; select "My Data" on the left bar, then select "Submit Data".

**3).** You will see the webpage "CREATE METADATA". Please fill in your data information, such as i) Overview (Title, abstract, data file naming, file size, time range,…),

ii) Reference, iii) Keyword(s),…etc. After complete, click "Save" to save the information.

**4).** Then select "Data Files". A new page will popup, where you will find

(i) The host ip address, (ii) the port number, (iii) the username, and (iv) the password to use for Filezilla.


**5).** On your local site, such as NCAR Cheyenne, open Filezilla at the directory where the data you would like to upload are located. Please use the information from (4) to remotely access the data center via Filezilla.

**6).** You will be at the root directory. The root directory is empty and you need to create a folder using the naming method mentioned in Section I, for example, UCLA_CFS-SSiB2 under





the ''root directory''. If you have created the folder before, you will find it, when you log back.

**7).** Then from your Filezilla window, you can drag your data from your local site to the newly created folder/subfolder, such as Task1.

**8).** Send an email to Duo at panxd@itpcas.ac.cn. Then she will synchronize the data for you directly!


**9).** Click "submit" to submit the online data in the window which appeared in step 3.

**10).** Duo will send you a confirmation email to confirm/acknowledge the proper submission. By that time, you should be able to see your data.

In case there is any problem/question, please contact Duo (panxd@itpcas.ac.cn) with cc to Ismaila (idiallo@ucla.edu) for help.

**B)** **Acquiring LS4P-I Project Data**

a) Log in to the online National Tibetan Plateau Data Center (http://data.tpdc.ac.cn/en/), using the aforementioned login details (see Section II).

b) Go to "LS4P_group" / "Personal Center"

c) Select "My Data", and then select "Review" or "My Draft"

d) You will see all the metadata belonging to LS4P group.

e) Under the metadata, click "edit" button, and move to "Data Files" item, you will find the host, port, username and passport for the specific group data you selected.

f) Open Filezilla using the information's from **e)**,

g) Now from Filezilla you can manage the LS4P directory and see what has been uploaded, along with the current directories/sub-directories.



**Data availability:**

The LS4P data are stored and will be distributed through the National Tibetan Plateau Data
Center (Li et al., 2020, http://data.tpdc.ac.cn/en/) and the U.S. Lawrence Livermore National
Laboratory (LLNL) Data Center Earth System Grid Federation (ESGF) node (Cinquini et al.,
2014,    https://esgf-node.llnl.gov/projects/esgf-llnl).    The    evaluation/reference    datasets    from
CAMS, CFSR, CMA, CRU, ERA-Intrim, MERRA-2, and NARR, as well as model data
discussed in this paper are archived at http://doi.org/10.5281/zenodo.4383284 (Xue and Diallo,
2020).

**Competing Interest**

The authors declare that they have no conflict of interest.

**Author Contribution**

Conceptualization: Xue, Zeng, Yao, Boone, and Lau; preparing the original draft: Xue; review
and editing of the manuscript: all coauthors. Authors are ordered by contribution, and those with
similar contributions are in alphabet order based on their last names.

**Acknowledgements**

LS4P is a project of the Global Energy and Water Cycle Experiment (GEWEX) Global
Atmospheric System Study (GASS) under the auspices of the World Climate Research
Programme (WCRP). We thank the support of the Pan-Third Pole Environment (Pan-TPE)
program (Grant No. XDA20100000) and the Second Tibetan Plateau Scientific Expedition and
Research (STEP) program (Grant No. 2019QZKK0200), and the U.S. National Science
Foundation (Grant No. AGS-1849654), and the U.S. DOE E3SM project at LLNL (contract No.
DE-AC52-07NA27344) in organizing and coordinating the LS4P activity.  Each LS4P-I model
group's efforts are supported by the participants' home institutions and/or funding agencies. We
also appreciate Professor Paul Dirmeyer of the Center for Ocean-Land-Atmosphere Studies, the
George Mason University, Dr. Thomas M. Smith of the National Environmental Satellite, Data,
and Information Service/NOAA, and Dr. Catalina Oaida's help in preparing this manuscript.



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





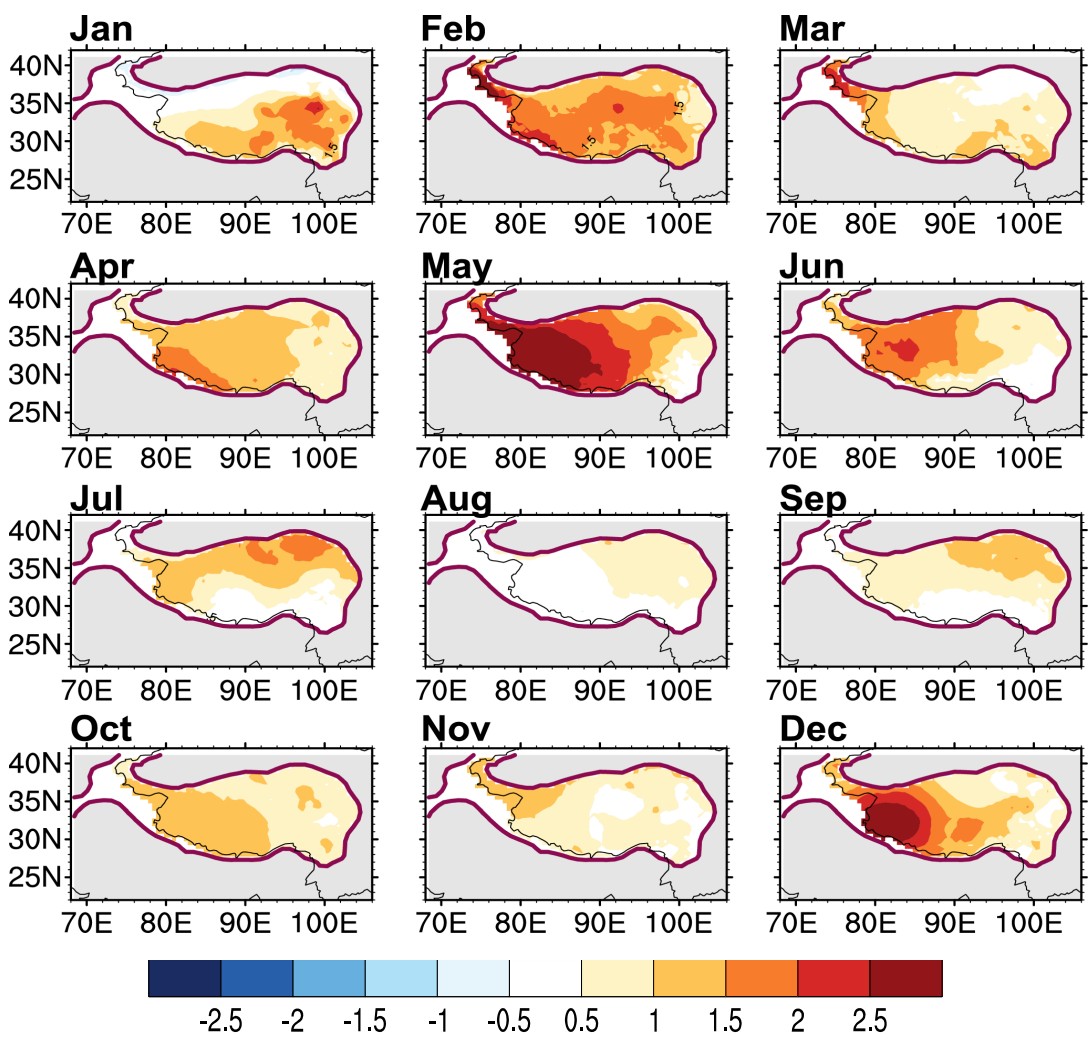

**Figure 1a. CMA Monthly 2-m Temperature difference between Warm and Cold Years (°C).**

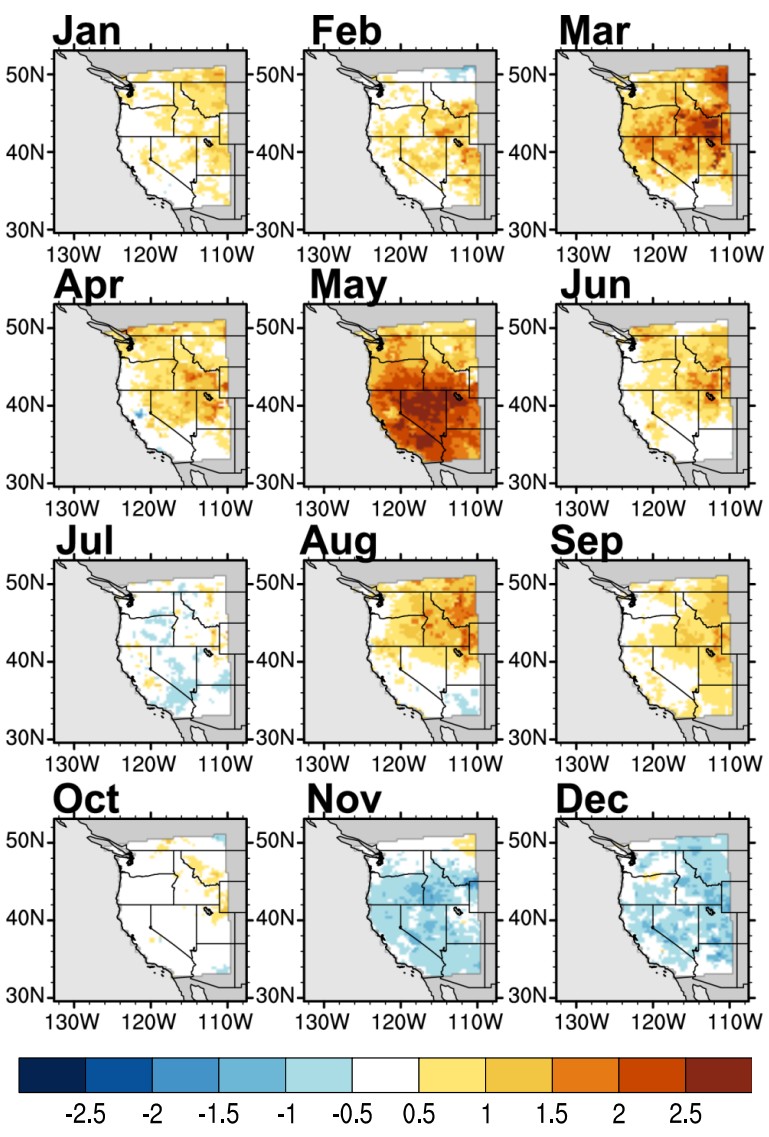

**Figure 1b. NARR Monthly 2-m Temperature difference between Warm and Cold Years (°C).**

Notes: **(1)** Years are selected based on the May anomalies using a threshold of one-half standard deviation during the period 1981-2010. **(2)** The North American Regional Reanalysis (NARR, Mesinger et al., 2006) assimilated the observed T-2m and is viewed as having an accurate representation of the observed surface air temperature.



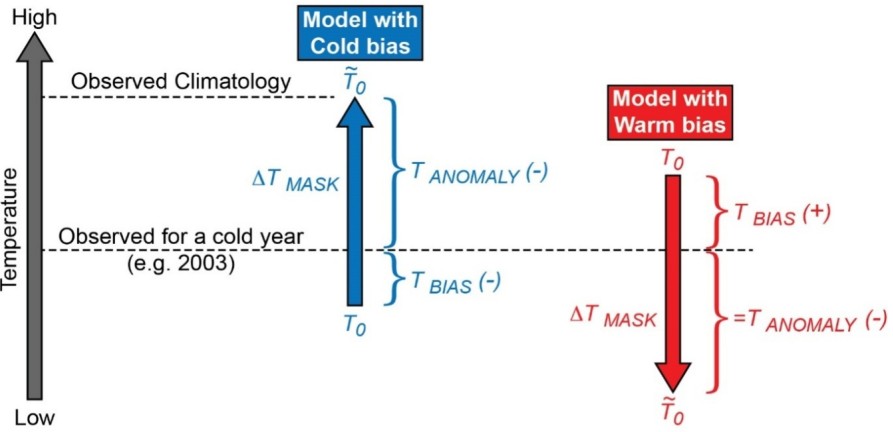

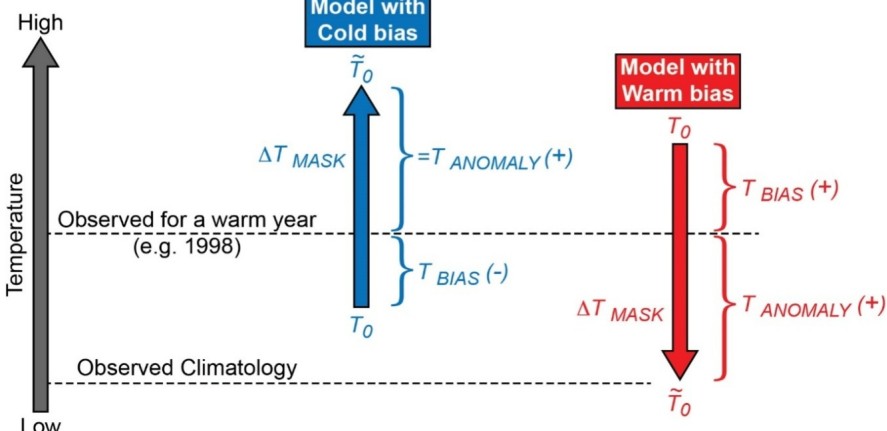

**Figure 2. Schematic Diagram for an Imposed Mask for Surface Temperature Initialization Corresponding to (a) a Cold Anomaly Year; (b) a Warm Anomaly Year**

Notes: 1). $T_0$ is the original model initial condition and $\widetilde{T}_0$ is the initial condition after imposing the mask. 2). The +/- sign in the parentheses indicate that the value is positive/negative, respectively.





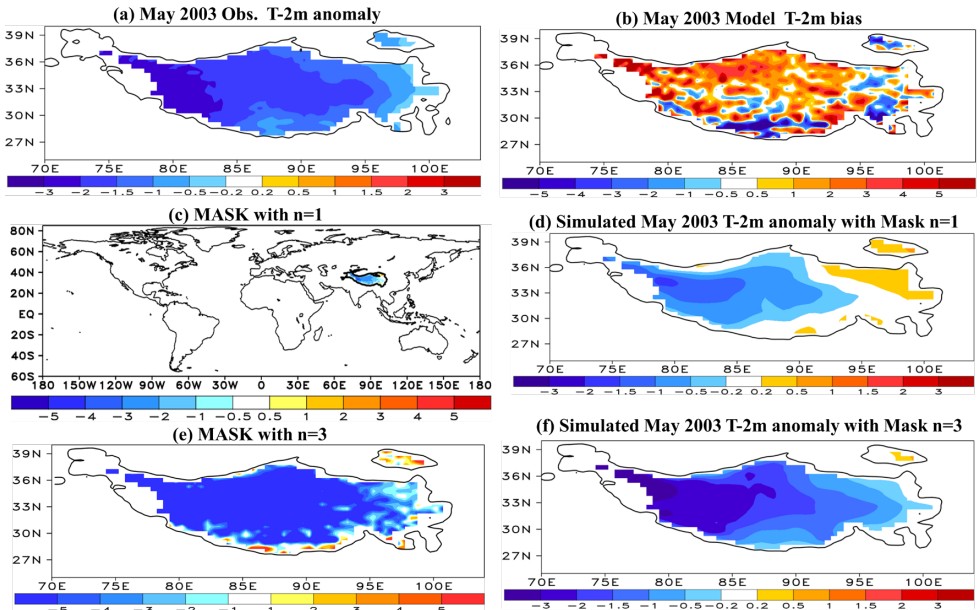

**Figure 3.** Schematic Diagram for the mask application; (a) Obs. May 2003 T-2m anomaly over the Tibetan Plateau (TP), (b) May 2003 T-2m simulation bias over the TP from a LS4P model, (c) Imposed mask with n=1 for a LS4P model, (d) Simulated May 2003 T-2m anomaly over the TP after imposing the mask shown in Fig. 3c, (e) as in Fig. 3c but with n=3 (only the TP is displayed), and (f) as in Fig. 3d but with n=3.

**Figure 4. May T-2m over the Tibetan Plateau above 4000m from observational and reanalysis datasets; (a) mean climatology, (b) 2003 anomaly, (a cold May) and (c) 1998 anomaly (a warm May).**

Note: The CMA climatology is used as reference for the anomalies. Because each T-2m data set has its own elevation, all the data have been adjusted to the CMA elevation for comparison.



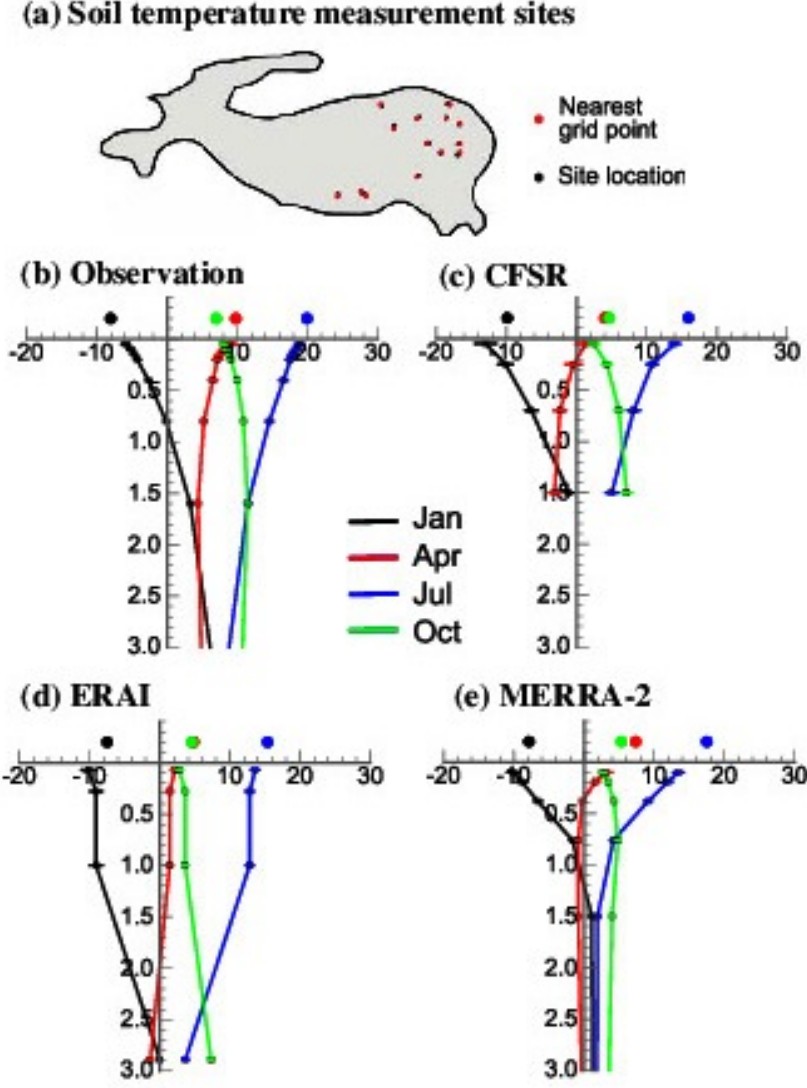

**Figure 5. Comparisons of the mean soil temperature profiles between the observational and reanalysis data in different seasons based on 14 Tibetan Plateau measurement site locations.**



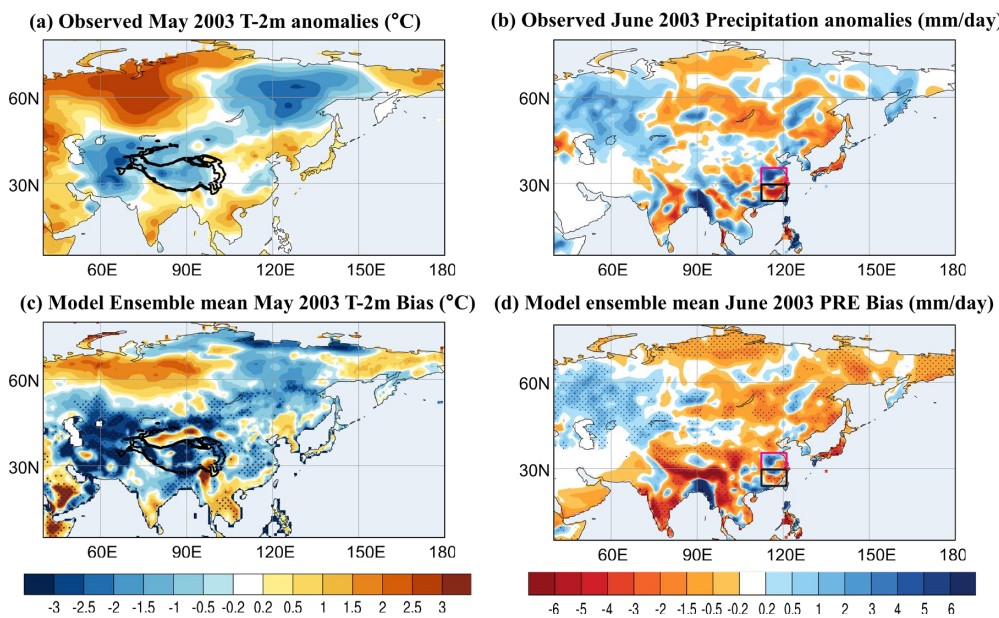

**Figure 6. Comparison between the observed anomalies and the ensemble mean bias for May 2003 from 13 LS4P-I Earth System Models (ESMs).**

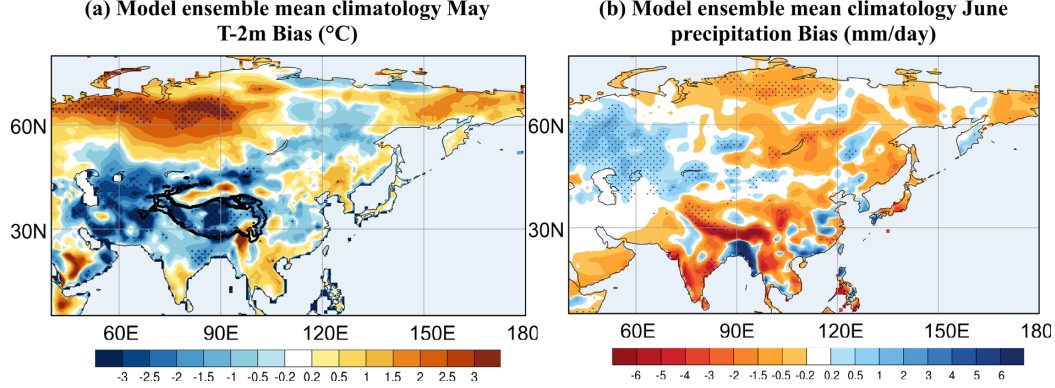

**Figure 7. Thirteen LS4P-I ESMs Ensemble Mean Climatology Bias.**