# Peer review of "Impact of Initialized Land Surface Temperature and Snowpack on Subseasonal to Seasonal Prediction Project, Phase I (LS4P-I): Organization and Experimental design"

_Geoscientific Model Development, 2020_

## Author Comment (AC1)

 Geoscientific Model Development Discussions Open Access **EGU**

Review of gmd-2020-329 by Xue et al.
"Impact of Initialized Land Surface Temperature and Snowpack on Subseasonal to Seasonal Prediction Project, Phase I (LS4P-I): Organization and Experimental design"

This paper introduces the LS4P project including its motivation, goals, and instructions on the modelling experiments. The key idea is to study the effect of soil temperatures in mountainous areas on subsequent precipitation in downstream areas through remote effects of land-atmosphere interactions. Models and observations hint that such effects could be existing, and could consequently be exploited for weather/climate forecasting. Anaccurate representation of these effects across models is challenging, and the project is aiming to improve this.

- - - - - - - - - - - -

Recommendation:
I think the paper requires major revisions.

This paper comprehensively presents the LS4P project which brings together the land- climate modelling community. The key idea about exploiting remote effects of land- atmosphere interactions for weather predictability is promising, and fits well with other recent studies illustrating so-far largely overlooked effects of land surface status and fluxes in downstream areas. Also considering soil temperature in this context is innovative.as it also reflects to some extent the moisture/energy state of the land surface, and quite some satellite and ground-based data are available which are partly insufficiently exploited, particularly in comparison with the more prominent soil moisture. Nevertheless,I also see some shortcomings in this paper which should to be addressed to make the paper suitable for publication in Geoscientific Model Development:

Response: The reviewer has very carefully reviewed the manuscript, and provided very insightful, constructive, critical, and encouraging comments and detailed suggestions. The manuscript has been revised based on the comments/suggestions. We sincerely appreciate the reviewer's efforts and have acknowledged the reviewer and editor in the revised manuscripts.

(1)
I have some doubts about the application of the mask as described in section 3.2.While equation 1a is clear, I do not understand why equation 1b is needed;
in this context also lines 403-408 and Figure 2 are unclear to me.
More generally, I think that the models' memory is a dynamic feature which should
be addressed through adapting the modelled (soil temperature and moisture) *dynamics*rather than their *states*.
This way, I feel if the tuning parameter "n" could exaggerate initial correctionswhich could degrade the simulations in the early forecasting period
Why not simply correcting the model/forecast biases through post-processing and without interferring with the actual model simulations?
Finally, as the mask correction requires observed soil temperature information, maybe I missed that but I was wondering how this is done in places where this is notavailable?

Response: The reviewer raises several issues for clarifications of our soil temperature initialization methodology. The following is our responses.

1). The LS4P project pursues a new approach, i.e., using the LST/SUBT anomaly in high mountain areas, to improve the S2S prediction. The current start-of-the-art models, however, are unable to

properly produce the observed anomalies, and then by extension, this anomaly-induced dynamic and the associated physical processes, in their simulations. As such processes are not existed in the model simulation; the bias correction in post-processing is unable to generate these processes. In fact, the LS4P deals with the S2S prediction, which is essentially the same as weather forecasting. A bias correction in the post-process is normally not employed.

2). It is a good idea to improve the model dynamic and physical processes to overcome the modeling problem. However, improving Earth system model/land model and reanalyses data (which problem is shown in Figure 4 of this paper) in order to overcome the deficiency in modeling high mountain land surface temperature anomaly is not a simple task and may take decades of effort (today's land temperature model development has more than 70-year history), but proper S2S prediction, including drought/flood prediction, is an urgent World Meteorological Organization (WMO) task with societal implications. On line 377 of the revised manuscript, we pointed out that preliminary research suggests that "prescribing both LST and SUBT initial anomalies based on the observed T-2m anomaly and model bias is the only way for the current ESMs to reproduce the observed May T-2m anomalies". This is current status in the LS4P modeling groups. Of course, we welcome any research group to find a new way to produce observed anomaly, but it is not a simple task. On other hand, using initialization to improve meteorological prediction is nothing new but a traditional meteorological approach. We believe when this issue gets more attention and more data from measurement are available, more methodologies may be developed. But the scientific development always takes time. We have to undertake any development by making step-by-step improvements.

3). The reviewer has raised question about the schematic diagram (Figure2), which is designed to help other modeling groups to reproduce the mask that we use to test our approach. We acknowledge that the presentations of the Figure 2, its caption, and relevant text were not comprehensive and had shortcomings in helping readers to understand our approach. We have had several iterations among our co-authors and revised the figure, the caption, and the related presentation in the text to help readers better understand our approach as well as the procedure to produce the mask for initialization. In particular, we have clarified in the revised Figure 2 which initial temperature is for Task 1 and which one is for Task 3 with more details and explanations in figure note. We also reorganize Section 3.2(3). We believe these should help readers to understand the overall ideas in this figure and in the LS4P-I experimental design.

Meanwhile, the original Figure 2 includes both warm and cold years. In the revised version, to make the thing simple and less confusion for readers who are unfamiliar with LS4P, we only include cold year (same as the case in the LS4P Phase I) in the text and move the schematic diagram for the warm case to appendix for readers who want to pursue this issue further for their own research.

4). Regarding the observational data, under the current big data system, it should be available very soon. The general public can obtain last month's observational data. In a major climate center, the real time analysis data are generated in a very timely manner. In terms of the model bias information, if it is unavailable in some cases, then the model climatology bias can be applied. Our paper (Section 5 and Figure 7) indicates that the bias is very consistent if comparing climatology and year 2003. The methodology does not require a precise bias value. But the general signs (positive or negative) are important.

5). The reviewer thought we may impose artificial large forcing because of a tuning parameter "n". It is not true. In the following paper in a Climate Dynamics special issue, we will show every model's-imposed forcing. They are not that extreme. The LS4P-I includes most of major climate centers in the world. If our approach is totally different from their normal practices, they would not endorse the LS4P-I and participate in this project. By the way, many parameterizations, such as convective schemes, aerosol parameterizations, etc. also have the

tuning parameters.

(2)
The description and motivation for selection of the ground truth data is insufficientin my opinion. The text in lines 249-273 lists many datasets but does not indicate
how they are applied. A summary table of all employed datasets would be nice, includingtheir spatial and temporal extent, advantages, and variables provided.
I understand that you are using the CMA dataset as this is based on a relatively largenumber of ground measurements. This makes sense, but it would be interesting how(well)
this dataset extrapolates between these measurements and how many (fewer?)measurements
are employed by other datasets such as the state-of-the-art ERA5 reanalysis. Next to this, I was wondering why you are not employing available satellite-basedland surface temperature products http://data.globtemperature.info/ ?

Response: The data sets described in this section are produced by the LS4P-I data group, and we intend to provide this information for scientists who are interesting in conducting further LS4P-I researches. These measurements are the foundation for the LS4P-I research. For instance, in the reviews of earlier papers/proposals of the LST/SUBT effect, some reviewers just use "there are no large scale SUBT measurements in high mountain areas to confirm the exist/presence of such anomaly" to suggest for rejection. Now the LS4P-I data groups have provided comprehensive data sets to support the community for the research. That's why we choose the Tibetan Plateau as the focus area for the LS4P Phase I and would like to provide relevant information for the community to introduce these data sets. In the revised manuscript, per the reviewer's comments, the role of the data group is clearly presented (lines 186-187; 251-254). We have also added a table in the appendix (Table B1) listing all the data that we used.

As to the station number in China, for other data sets, such as ECMWF data set, they normally obtain the data from GTS system and through collaborative agreements. Normally, they have less than 400 stations, much less than what we have listed in the text.

Remote sensing over the Tibetan Plateau is challenging because of a lack of validation data (for testing the algorithms). Since the Tibetan Plateau is one of the focus areas of GLASS satellite products, the GLASS group has more experience on this area's remote sensing with better quality there. In addition, the GLASS group is participating in the LS4P-I project. That's why we mainly use the GLASS products in the project. But we certainly do not exclude other satellite products and will use them if they can provide useful information for the project.

Thank you for sharing the link (http://data.globtemperature.info/) for surface temperature data. Unfortunately, all these datasets have a short coverage period and do not cover the climatology period (1981-2010 and 1980-2013), which we have considered in our study.
(3)
The manuscript is comprehensive but also quite long. To make it more concise it would behelpful to shorten where possible I think. Below, I have indicated some examples where content
is repeated and where I would see potential to shorten the text.
Moreover, a summary table listing the main information regarding tasks 1-5 in section 3.2would improve the readability.
Response: Thanks for the reviewer's suggestion. We will make the revision according to every suggestion that you list in specific comments (see our response in Specific Comments below for detail). In the revised manuscript we have added a table (Table 1) to clearly provide information's regarding Tasks 1-4 as suggested.

(4)
The authors refer a lot to snow effects in sections 1 and 2, and I like these ideas.However, snow is not mentioned at all in sections 3, 4 and 5, and apparently only implicitly (through LST) part of

the analyses.
This should be clarified, and the role of snow in the project as described insections 1 and 2 should be toned down.

Response:  The snow part of the work is an important component in LS4P. Snow is one of the major driver to produce LST/SUBT anomalies.  That's why it is included in the project's name.  In Phase 1, however, we are mainly looking for first order effects most related to the soil surface and deeper layers; but indeed, we mentioned snow effects as we (agree with this reviewer) think it is important and related to the LST/SUBT anomaly, and plan to examine it in the Phase II.  Since this is the main paper introducing the LS4P, we have to present the importance of snow in this project in section 1.  Otherwise, the readers may immediately raise issue about where the LST/SUBT anomalies come from.
        The shortcoming in our previous presentation was not to provide a clear expectation how much snow related activity will be discussed in this paper, and make some readers keep waiting till the end of the paper.  We now make a clarification at the end of Section II that this part of research will be considered in Phase II papers and will not be presented further.  So readers (who are interest in this) will have a proper expectation.

(5)
As with the snow, also predictability is prominently mentioned in sections 1 and 2 and even in the abstract and title, but the detailed description of the project and thesimulations
does not refer at all to this. So also here I would suggest to either include details
on how the predictability could/will be assessed, or tone it down in the beginning of themanuscript.
Response:  Please see our response above.
I do not wish to remain anonymous - René Orth.
Response:  Thanks.  We have acknowledged the reviewer in the revised manuscript.

‒ ‒ ‒ ‒ ‒ ‒ ‒ ‒ ‒ ‒ ‒ ‒.

Specific comments:
line 29 & 179: "Data groups" is not clear.
Response: Sorry, we failed to clearly provide the relevant information. In the response to your main comment 2, we had a discussion on this issue.  In the revised paper, we have clarified their contribution to the LS4P on lines 186-187; 251-254.

line 35: Summer precipitation in which area?
Response: We have added "beyond East Asia" on line 35

line 49: "stubbornly low", not everywhere, there are quite some regional variations ofprecipitation forecast skill
Response: The S2S prediction is associated with drought/flood/heatwave prediction. In N. America, Europe, West Africa, and East and South Asia, the Earth System Models have difficult to reproduce these events.  That's why the WCRP and the WRRP/WMO list S2S as the current high priority.   In some areas, such as California, the weather and climate have low variations in some seasons.  But this is not the focus of our S2S prediction research.

lines 60-80: in this context you could cite Orth and Seneviratne (2017) where we comparethe impacts of SST vs soil moisture
for land climate globally
Response:  Added and thanks for the suggestion

lines 83-85, and elsewhere: I am missing some justification why you chose to focus on (usually data sparse) mountain areas, and why their downstream impacts are expected tobe higher than that

of flat regions with possibly larger surface heat fluxes.

Response: Lines 85-111 (the revised version) in the introduction and the 1st paragraph in Section 2 discuss the justification to use high mountain areas as focus for S2S prediction. The LST's effect in high mountains have been long overlooked until scientists in the LS4P groups conducted preliminary researched discovering its important role in S2S predictability. The preliminary sensitivity experiments and relevant papers presented in this part show the effect of spring LST/SUBT in high mountain areas on downstream summer precipitation. The papers cited here also discussed the mechanisms why the impacts are in downstream. Because of the high elevation, the perturbations induced from the land surface process could propagate for long distances through Rossby wave interactions affecting the downstream areas. We have modified the paper in the introduction to help the reader to better relate the discussions there to the justification as to why we choose a high mountain area as the focus.

Yes, the mountain area measurements are overlooked because of the difficulty making measurements in the environmental condition there and because of failing to understand the importance of the measurement in these areas. We hope our research will stimulate more measurements in these areas.

line 86: "very close", you mean strongly related here?
Response: On line 89 of the revised version, the text has been modified to clarify the meaning here.

line 89 & Figure 1 caption: I do not fully understand how this was computed. Do youselect years with warm/cold Mays, respectively, and then you track the temperaturedifference between these years through all months?
Response: Yes, we use the approach as you mentioned. This approach has been pointed out in the revised note after the caption of figure 1. We further emphasize it on line 95 of the revision

line 100: SSiB, abbreviation not explained
Response: Added

line 122: insert "the" before "snow darkening effect"
Response: done

lines 127/128: could you please specify/explain "large diversity" a bit more?
Response: we have modified the sentence and in the revised version of the manuscript it reads as "this could be one of the major reasons for the large discrepancies in simulated T-2m and its anomaly in current Earth System Models (ESMs)".

Line 146: insert "the initiative" after "historical development of"
added

lines 178/179: there is no need to put both written-out and numeric numbers there
Response: agree. We have eliminated the written-out numbers.

lines 195/196: I think this is an important part of the project, can you give some moredetails on this data base?
Response. We have added relevant information for the data base on line 203 of the revised version.
lines 210/211: Sorry for mentioning an own study again, but Orth and Seneviratne (2017)can be instructive here I think
 Here just summarize questions, no any citation listed here. The relevant citations are in Introduction and the reviewer's paper has been added in citation there.
lines 221-225: repetition, could be removed line 246: insert "the" before "Tibetan Plateau"
done

line 248: please be more specific which of these datasets are useful for the project, andwhy
The data used for this project is clarified on lines 249-251 with a table in Appendix B in the revised version. This section mainly presents data sets produced by the LS4P data groups. Only some of these data are used by the LS4P phase I experiment. Most data introduced here can be used for the LS4P related research, such as the causes of LST/SUBT anomalies, Tibetan Plateau land surface characteristics (for instance, snow, frozen soil) associated with land memory and land surface energy and water balances, and land/atmosphere interaction there. The multi-model testing in the LS4P project can only investigate some key issues and intends to stimulate more research on this aspect from the community, which is needed to ultimately understand this issue. This section provides the relevant data information for the community, which is very useful for them to conduct high mountain-related researches. We have clarified this in the revised section 3.1 of the manuscript.

lines 267-269: could be removed
As indicated earlier, during the early LST/SUBT research, this was the precise reason some reviewers reject our approach. We make the statement here is to show although some of the data do not directly use in LS4P phase I experiment, but they do provide basic information/evidence to support the crucial justification for the LS4P activity.
lines 282-283: you mean anomaly precipitation rates of +1.32 mm/day?
Response: Yes. Anomaly is added.

line 294: "around late April", why is this not more specific?
Response: The LS4P requests for at least 6 members for each Task, which normally means 6 different starting dates (for initial conditions), such as April 25, April 26, April 27, April 28, April 29, and April 30. Every modeling group may select different days based on their normal practice for numerical prediction (for instance, some days from April 26-May 1, and some from April 28-May 3). That's why we use "around late April" here.

line 316: "the models' performances are then checked" can be removed
done

line 321: what is a "proper" lapse rate?
Response: This issue is discussed in Xue et al. (1996) and Gao et al. (2017). We modify the sentence to make it clear.

lines 323-327: repetition, can be removed
Okay.

lines 334-335: Not sure if the project is obsolete if models would do a good job, as you could still study LST/SUBT downstream effects, as stated in the project goals in e.g. the abstract
Response. The downstream effect is a new scientific discovery for the S2S predictability regardless whether the models are able to produce the LST anomalies on the mountain areas. However, if the models were able to produce proper temperature anomaly over the Tibetan Plateau, the focus would not be how to generate the observed LST anomaly. We have modified the statement on line 344 and add lines 624-625 to make it more adequate and precise.

line 349: "around 2010", why is this not more specific?
Response: Every climate modeling center has made long term climatological runs but with slightly different starting and ending years. We are unable to request these big centers to redo their climatological simulation because of the huge amount of work load and computer time, so as a best approximation we only ask them to send their climatology which is an average around certain time period in order to be compatible with our experiment. We believe a climate based on the average 1980-2010 and another from 1979 to 2009 should have no fundamental difference.

line 426: what are the "sensitivity" and "control" runs?
Response: The sensitivity is Task 3 run and control is Task 1 run. We have clarified this.

line 450: here you could point to Table S1
Done. Thanks.

lines 478-479: ERA-Intermin should be updated to ERA5 (https://climate.copernicus.eu/climate-reanalysis)
We did not update due to the time period that ERA5 covers. The GMD also does not allow to put website address in the paper, so we still use the previous citation.

lines 473-475: I get the point that you would use the dataset that uses most ground station measurements as reference, but I am wondering how many stations ERA5 is using over this area?
Response: I do not know exactly how many stations ERA5 has. But based on my information (pers. Comm.) it should not be more than 400 stations in China, which is much less than what we use.

lines 590-591: While it becomes more clear later, it would be good to already motivate here why you are comparing biases with anomalies.

Response. This is a good suggestion. We have added lines 624-625 to indicate why we want to compare bias and anomalies.

line 594 & Figure 6: I do not understand why you do this multiplication with -1.
Response: Results in Figure 6 provide the underpinnings for the LS4P conjecture: if the May land temperature anomaly on the Tibetan Plateau does contribute to the June East Asian precipitation anomaly, then improving the May land surface temperature simulation over the Tibetan Plateau through an improved initialization should make Earth System models to produce better June East Asian precipitation.

For instance, May 2003 was a cold month for the Tibetan Plateau and June 2003 was dry to the south of the Yangtze River in the observations. If we postulate the May cold T2m in the Tibetan Plateau caused the drought, then a model with cold (warm) bias in May in the Tibetan Plateau should produce a dry (wet) bias to the south of the Yangtze river if these Earth System models' dynamics and physics reflect such linkage that in the real world.

Because some models have a cold and dry bias and some have a warm and wet bias, when we make the composite in Figure 6, we have to multiply "-1" for the models with warm/wet bias to integrate them with the models with cold and dry bias (to avoid their biases cancel each other), and to compare them with observed anomalies.

line 605: How is this correlation computed? Is it a spatial correlation? Over which domain?
Response: it is a spatial correlation over the figure domain, which now is specified in the revised version (lines 638-639).

line 615: Please specify the area over which the subsequent drought occurred.
It is clarified on lines 647-649

line 637: Why these methods, and not a similar approach as for example in Koster et al. (2016)
Response: Here we deal with the prediction. Smith et al.'s work on this line of the original manuscript is a statistical prediction. Koster et al.'s work (2016) conducted series of stationary wave model (SWM) experiments in which the boreal summer atmosphere is forced, over a number of locations in the continental United States, with an idealized diabatic heating anomaly. As such, Koster's work is an ideal sensitivity study and is different from what we pursue

(prediction study).

lines 643-647: Why and how is the fore-restore method causing inaccurate soil memory?
Response: We have two published studies (Liu et al., 2020; Li et al., 2021) addressing how the force restore method causes the problem and how to improve the memory through the proper parameterizations. Since a comprehensive discussion on these issues are out of the scope for this paper, in the revised text we added these two citations on line 680.

Liu Y., Y. Xue, Q. Li, D. Lettenmaier, and P. Zhao, 2020: Investigation of the variability of near-surface temperature anomaly and its causes over the Tibetan Plateau. J. Geophy. Res. 125, e2020JD032800. https://doi.org/10.1029/2020JD032800.
Li, Q., Xue, Y., and Liu, Y.: Impact of frozen soil processes on soil thermal characteristics at seasonal to decadal scales over the Tibetan Plateau and North China, Hydrol. Earth Syst. Sci., 25, 2089–2107, https://doi.org/10.5194/hess-25-2089-2021, 2021.

line 666: Correct tense, 2020 is in the past now :)
Corrected

line 669-670: "A possible ... will also be prepared" can be remove
Done

Figure 5:
- the quality/resolution is very low, please improve
Done

References:

Orth, R. and S.I. Seneviratne
Variability of Soil Moisture and Sea Surface Temperatures Similarly Important for Warm-Season Land Climate in the Community Earth System Model
Clim. Dyn. 30(6), 2141–2162 (2017).

Koster, R., et al.
Impacts of Local Soil Moisture Anomalies on the Atmospheric Circulation and on RemoteSurface Meteorological Fields during Boreal Summer: A Comprehensive Analysis over North America
J. Climate 29(20), 7345-7364 (2016).

---

## Author Comment (AC2)

Geosci. Model Dev. Discuss., referee comment RC2
https://doi.org/10.5194/gmd-2020-329-RC2, 2021
![CC BY]

**Comment on gmd-2020-329**

Anonymous Referee #2
* * *
Referee comment on "Impact of Initialized Land Surface Temperature and Snowpack on Subseasonal to Seasonal Prediction Project, Phase I (LS4P-I): Organization and Experimental design" by Yongkang Xue et al., Geosci. Model Dev. Discuss., https://doi.org/10.5194/gmd-2020-329-RC2, 2021
* * *
This paper describes the international / multi-organization LS4P project and provides some initial analyses of data submissions. It is essentially an "introduce the community to the project" paper, with most of the scientific findings to be documented later, as the project progresses.

While the paper is well written, in my opinion it has enough issues to rate a review of "major revision". On the one hand, I do applaud the experiment organizers for bringing together such a wide-ranging group of participants to address such an important problem. With this diverse a group and a corresponding collection of model outputs from such a diverse set of models, I don't doubt that the project will bear useful scientific fruit, and a paper like this that introduces the project to the broader community is certainly of value. On the other hand, the paper's write-up glosses over several critical aspects of seasonal and subseasonal prediction that call some of the project's long-term strategies into question, at least in terms of how they're currently described. A revised version should address these shortcomings through substantial qualification (not just a sentence or two here and there) or, better yet, through a substantial rethinking of the approaches to be applied.

Response:  We thank for the reviewer's acknowledgment that we have a wide-ranging group of participants to address an important problem and that the project will bear scientific fruit. We have taken into careful considerations of the reviewer's comments/concerns on our writing and questioning of our strategies.  The following is our point-by-point responses to address the issues that the reviewer raises.  We believe some issues are rooted in the reviewers understanding in our experimental design, thus in our revisions, we have made a great deal of effort to improve the presentation of our experimental design section.  The following responses explain and address the reviewer's questions.

1. The underlying assumption of the project appears to be that if a model does not produce an accurate temperature over the Tibetan Plateau, the fault lies with the land model (e.g., in how long the land model maintains an initial condition). The paper states this explicitly on line 465. The truth is, all models have biases in both air temperature and precipitation across the globe, and these biases could have any number of sources. A temperature bias over the Tibetan Plateau might have nothing to do with land model processes. It might instead result from deficiencies in the reproduction of the general circulation, for example, or from some problems with the radiation balance. Forcing the model to have a low temperature bias by imposing a stronger initial temperature anomaly (perhaps even an unrealistic anomaly, through eq. 1) may amount to "getting the right answer for the wrong reason", which is not a good basis for a forecast experiment. It's

quite possible that forcing a correct temperature through such an initialization when the model wants to do something else for reasons unrelated to land processes might have unexpected negative consequences – especially if the model is artificially modified in one region and not in surrounding regions. Substantial discussion regarding this is needed.

Response: we fully agree with the reviewer's comments that "all models have biases in both air temperature and precipitation across the globe, and these biases could have any number of sources," and when you correct the bias you may be "getting the right answer for the wrong reason".

But we have to clarify a few issues:
(1). Numerous modeling studies, since the very beginning of meteorological model development, have worked on correcting model-produced precipitation and temperature errors by improving some model parameterizations and through improving initial and boundary conditions. Our approaches and statements are based on a number of published papers in our field's major journals. We cannot simply speculate these peer-reviewed research's results are produced due to the wrong reason unless there is an evidence to support such statement. Otherwise, we may eliminate any modeling improvement studies because, at least at its early stage, they cannot prove that improvement is not due to a wrong reason. Only the community collective efforts with long term exercise can prove it.

(2). The reviewer speculates the problem in simulating the LST anomaly may not be due to the surface models. In general, when we try to correct a model deficiency in one variable, it is normal to check the dynamic and physical processes relevant to this variable first. For instance, for the precipitation simulation errors, we naturally check the convective and cloud process modeling and surface evaporation parametrization first. Unless there is an evidence to show these are other reasons, we cannot claim it is a wrong approach. Our statement on land model deficiencies are based on published papers and analyses from the LS4P research. On lines 126-129, we present a publication (Liu et al., 2020), which focuses on exploring the causes of model deficiency in properly producing the observed surface temperature anomaly in high mountains. That study demonstrated this deficiency IS associated with the land surface process model, including snow/albedo and soil subsurface memory effects. Recently, we had another published paper (Li et al., 2021) address this issue and further confirms Liu et al.'s conclusion. In addition, we have also analyzed the reanalysis data, which are used for model initialization, and indicated that the deficiencies in reanalysis data in high mountain areas also contributes to the simulated surface temperature bias. We believe we have caught the main (if not all of the) reasons for the model deficiency for this aspect. We have more clearly indicated our statements are based on the published papers in abstract and conclusion, and welcome more research to explore this issue in the revised version (Lines 493-496 )

Liu Y., Y. Xue, Q. Li, D. Lettenmaier, and P. Zhao, 2020: Investigation of the variability of near-surface temperature anomaly and its causes over the Tibetan Plateau. J. Geophy. Res. 125, e2020JD032800. https://doi.org/10.1029/2020JD032800.
Li, Q., Xue, Y., and Liu, Y.: Impact of frozen soil processes on soil thermal characteristics at seasonal to decadal scales over the Tibetan Plateau and North China, Hydrol. Earth Syst. Sci., 25, 2089–2107, https://doi.org/10.5194/hess-25-2089-2021, 2021.

(3). However, to improve land model and reanalyses data for this aspect are not simple tasks and may take decadal effort (today's land temperature model development has a more than 70-year history), but proper S2S prediction, including drought/flood/heat wave prediction, is a WMO task requiring urgent attention owing to a significant societal demand. On line 377-379, based on the LS4P modeling group's practice, we have pointed out that preliminary research suggests "prescribing both LST and SUBT initial anomalies based on the observed T-2m anomaly and model bias is the only way for the current ESMs to accurately produce the observed May T-2m anomalies". In fact, using initialization to improve meteorological prediction is nothing new but a traditional meteorological approach. We believe when this issue gets more attention and more data are/become available, more methodologies may be developed to address this issue. But

the scientific development always takes time. We have to allow any development to take step-by-step improvement.

(4). We agree with the reviewer that the statement on line 465(previous version) may be misleading and therefore we have modified the text to more properly reflect the ideas that we present in this paper (Lines 493-501).

2. The overall strategy seems to ignore the fact that forecast models generate their forecasts relative to their own climatologies. A model that is known to be biased warm may produce an anomalously cold 2003 over the Tibetan Plateau (compared to what it usually produces there), and that would be useful information even if the forecasted temperatures are still warmer than the average observed TP temperature. The point is that people know how to account for long-term model biases. They would properly consider a forecast model's result to be "2003 will be colder than usual by 5 degrees" rather than "the temperature will be 20C". The emphasis here on matching the observed temperature in absolute magnitude seems inappropriate to a discussion of forecast systems. See, e.g., the NMME forecast anomaly pages [https://www.cpc.ncep.noaa.gov/products/NMME/seasanom.shtml], which show forecasted anomalies relative to each model's climatology.

Response: The reviewer raises several issues here.

(1). Because of the model systematic bias, some groups indeed applied the anomaly prediction in their normal practice. We understand the justification for these groups, including some LS4P groups, used the anomaly predictions, but predicting the temperature in the real world is always our ultimate goal because the public needs the forecast for the real world, not the forecast relative to one group's model climatology.

(2). However, in the LS4P experimental design, as the first step, we only intend to see if there is any relationship between the observed Tibetan Plateau spring LST/SUBT anomaly and downstream summer precipitation anomalies. We mainly look at the difference between Task 3 and Task1. In this way, the model systematic bias has been eliminated so what we really look is indeed the anomaly, which is just what the reviewer tries to emphasize here. Because of this approach, our goal here does not emphasize producing the best initialization for May 2003 per say, although the methodology present here should serve this purpose. We have made major revision for Section 3.2
(3). In the modified manuscript, we have clarified our idea/approach.

We acknowledge that our schematic diagram and relevant text do not emphasize this idea clearly as pointed out in the Reviewer's third minor comment. We have modified the schematic diagram (Figure 2), figure captions, and reorganize the "Section 3.2(3) Task 3" in the revised manuscript to make the idea clearer. Original Figure 2 includes both warm and cold years. In the revised version, to make the thing simple and less confusion for readers who are unfamiliar with LS4P, we only include cold year (same as the case in the LS4P Phase I) in the text and move the schematic diagram for the warm case to appendix for readers who want to pursue this issue further for their own research. In our response to the reviewer' 3[rd] minor comment below, we will have some more detailed explanation.

3. Forecast systems also produce a range of forecasted values through the running of ensembles, and any one ensemble member could represent what happens in nature. The experimental analysis protocol, however, emphasizes the importance of having the *ensemble mean* match the observed anomaly. This is inappropriate. The key question is, do any of the ensemble members look like the observations? (And, in conjunction with point #2 above, the truly key question is, do any of the ensemble member anomalies *relative to the forecast model's climatology* look like the observed anomaly?) A model cannot be considered wrong if one of its ensemble members looks like the observations. Insisting that an ensemble mean match a specific year's temperature seems wrong.

Response: In this paper, we do not emphasize the model intercomparison as well as each model's evaluation because the major focus for the LS4P is whether the LST /SUBT can provide S2S predictability through the multi-model efforts, which idea has never been tested before.  Since many multi-model projects, such as CMIP, WAMME, and many others, find the ensemble means normally produce better results than any individual model's performance, we use the ensemble mean and the range of individual model results to assess whether there is S2S predictability and its uncertainty.  The reviewer may have different opinion on this, but this is a common approach currently used in multi-model studies in the community, such as CMIP, AMIP, WAMME and endorsed by the LS4P modeling groups. In addition, the LS4P has more than 20 ESMs and many RCMs.  A comprehensive analysis of each model performance is not that closely related to our main focus at this stage.

4. The model results concerning May Tibetan Plateau temperature anomalies and June precipitation anomalies in east Asia is perhaps suggestive but far from indicative of a causal relationship. Even if the agreements in 6a/6c and 6b/6d do suggest that one pattern led to the other (it could very well be coincidental), I don't see how the Tibetan Plateau in 6a/6c can be isolated as the source of the agreement in 6b/6d. Significant qualification of this figure's implications is needed.

Response.  We fully agree with the reviewer's comments and apologize not to have presented our ideas more clearly.  Figures 6a/6c and 6b/6d are NOT intended to present the causal relationship. These figures only intend to explain how we develop our hypothesis.

Observational results in Figure 6A, B (from various observational data sets) along with the remarkable consistency of modeling results in Figure 6C, D (compared to Fig. 6A, B) together provided the underpinnings for the **LS4P conjecture** that if the May land surface temperature anomaly on the Tibetan Plateau does contribute to the June East Asian precipitation anomaly, then improving the May land surface temperature simulation over the Tibetan Plateau through an improved initialization should allow Earth System models to better predict June East Asian precipitation.

The scientific development normally starts from scientists' curiosity based on some preliminary discoveries.  The reviewer apparently is an expert in the Earth system modeling and should understand such consistency in Figure 6 from various observational data sets and various Earth system models (with very different dynamic processes and physical parameterizations) is not by chance, and is worth to **propose a hypothesis** then explore this issue further.

By and large, Figure 6 is an important step to motivate the hypothesis.  Only Task 3 (with Task 1) is designed to prove the casual relationship.  We have more explicitly emphasize this point in the revised paper to avoid confusion (Lines 349-353, 624-625).

Minor comments:
-- Just to clarify: Are the warm and cold years the same for each month shown? If not, it's not clear what this figure says about the persistence of warm and cold anomalies (line 92).
Response:  Yes.  The years are the same for each month.  We have clarified this in the Figure 1 caption of the revised paper.

-- lines 106-108. This sounds strange. Can the authors clarify the link between temperature and water amount? While I see that more water leads to a slower seasonal transition, it's not obvious that at a single point in time, temperature tells you something about water amount.
Response: This part (lines 112-115 in the revised version) has been revised in the revision. The word linking temperature and water amount is indeed potentially confusing and has been eliminated.

-- I studied Figure 2 for a long time and still can't make sense of it. Why, for a cold year initialization, does the initial condition for a model with a cold bias get set to climatology whereas that with a warm bias does not? Also, please clarify in the caption: are the biases

discussed here errors for the particular year of simulation, or are they long term climatological biases? I'm guessing the former, since Task 2 would need to be done for the latter; in that case, though, the use of the term "bias" is confusing here. Bias should refer to a long-term climatological error (reflecting a model deficiency), not to the error at a specific time (which should reflect both bias and random error). Overall, Figure 2 is not helpful for explaining the approach. And again, based on my earlier comments, I'm not convinced the Tmask strategy is appropriate anyway.

Response:  This paper is for the LS4P experimental design and Figure 2 shows how we tried to generate the observed T-2m anomalies using the imposed mask in initialization then checked whether this anomaly improves the S2S predictability and leads to better prediction of the observed drought and flood events.  Based on the reviewer's comments, we recognize that we did not explain the idea clearly in the previous figure, figure caption, and text.  Although the researchers (co-authors of this paper) working in the LS4P know the idea from Figure, but probably not the readers who are unfamiliar with the project.  We have performed several iterations within co-authors to improve the figure, figure caption, and presentation in the text.  The section 3.2(3) is reorganized.  We hope that the revised figure and text clear-up the confusion.

In the revised figure, we have clarified which initial temperature is for Task 1 and which is for Task 3 with more explanations and indicate that the LS4P phase I's goal is to prove the causal relationship between the Tibetan Plateau spring LST/SUBT anomaly and large-scale summer precipitation anomaly. Figure 2 and Equation 1 show how to produce observed surface temperature anomaly through Task 3 initialization (relative to Task 1's initial condition).  We believe these should help readers to understand the ideas better in this figure.  With the revised figure, the readers can see that when we use the difference between Task 3 and Task 1, we actually try to avoid the model systematic bias which was precisely suggested by the Reviewer in the main comments.

Regarding the bias, here we did not clearly indicate whether this should be a specific year or a climatology because it depends on case and data availability.  "Bias" is not always associated with climatology. As Pan et al (2001, JGR) state in their analysis that "Both GCM and RCM fields can exhibit substantial systematic differences from gridded observational data. Such discrepancies between simulated and observed fields are commonly referred to as biases", although "some differences clearly are not biases in the strict sense, but for simplicity we use the term "bias" to refer to the entire set of comparisons".  Such interchanges are also used in other studies/fields.  A recent paper "Precipitation Biases in the ECMWF Integrated Forecasting System" by Lavers et al (2021) discusses using the IFS control forecast from 12 June 2019 to 11 June 2020 to show that in each of the boreal winter and summer half years, the IFS has an average global wet bias".  The way they use bias is similar to what we use.  In remote sensing, "bias correction" terminology is also commonly used.  Moreover, as discussed in the paper, from Task 2, we know the climatological T-2m bias and year 2003 T-2m bias are very consistent.    Therefore, we point out this terminology issue but still keep "bias' in our paper for simplicity as did in other current practices on lines 362-365.

-- Equation 1 appears to be a means to impose an artificially large temperature anomaly at the start of a simulation so that the anomaly is maintained realistically during the forecast. As far as I can see, there's no physical basis for the equation; it's fully empirical and could lead to initial temperatures that make little physical sense (e.g., colder than the model ever gets). More qualification is needed regarding how artificial this construct is. (And again, based on my earlier comments, it may not be appropriate to fix the temperature error in this way, since it may have a source other than the land model.)

Response:  The LST/SUBT approach is a new development and is at a very early stage.  The importance of memory of surface temperature has still not been fully recognized by the community.  Currently, no ESMs, including reanalyses, are capable to reproduce observed high mountain T-2m anomaly.  Developing adequate numerical methods and physical parameterizations to permanently solve the issue may take decades or longer.  In meteorology, using the initialization scheme before we develop better models and better data sets to improve the prediction is a very common approach, as done by Yeh et

al. (1984, MWR) and Yang et al (1994 MWR).  Those initialization strategies were always based on some empirical relations and not a strict physically-based approach.  Especially in the early stages, some approaches are highly idealized.  For instance, in Koster at al. (2004), which is a rather famous study, were used an approach for which a soil moisture value is artificially imposed for every time step.  But that did not prevent that paper from receiving more than 2250 citations and from becoming a classical paper in meteorology.

On other hand, our approach is not that extreme.  The reviewer thought we may impose an artificially large forcing because of a tuning parameter.  In fact, this is not the case. In the follow-up paper in a Climate Dynamics special issue, we will show every model's-imposed forcing.   in fact, they are not extreme.  The LS4P includes most of major climate centers in the world.  If our approach is totally different from their normal practices, they would not endorse the LS4P and participate in this project.

If we wait until the best dynamic and physical method are developed, nothing will happen.

Reference

Koster, R. D., P. A. Dirmeyer, Z. Guo, G. Bonan, E. Chan, P. Cox, C. T., Gordon, S. Kanae, E. Kowalczyk, D. Lawrence, P. Liu, C.-H. Lu, S. Malyshev, B. McAvaney,K. Mitchell, D. Mocko, T. Oki, K.. Oleson,  A. Pitman, Y. C. Sud, C. M. Taylor, D. Verseghy, R.Vasic, Y. Xue, T. Yamada, 2004: Regions of strong coupling between soil moisture and precipitation. Science, 305, 1138-1140.

Lavers, D. A., S. Harrigan, and C. Prudhomme, 2021: Precipitation Biases in the ECMWF Integrated Forecasting System, JHM, 22, 1187-1198.  DOI: https://doi.org/10.1175/JHM-D-20-0308.1

Pan, Z., J. H. Christensen, R. W. Arritt, W. J. Gutowski Jr., E. S. Takle, and F. Otieno, 2001: Evaluation of uncertainties in regional climate change simulations. J. Geophys. Res., 106, 17 735–17 751, https://doi.org/10.1029/2001JD900193.

Yang, R., M.J. Fennessy and J. Shukla, 1994: The influence of initial soil wetness on medium range surface weather forecasts, Mon. Wea. Rev., 122, 471-485.

YEH, T.-C., WETHERALD, R.T. and MANABE, S. (1984). The effect of soil moisture on the short-term climate and hydrology change—A numerical experiment. Mon.Wea.Rev. 112; 474-490

-- line 211 (and elsewhere): Replace "SST" with "ocean state", since SST is only a small part of what seasonal forecast systems rely on from the ocean. Arguably, subsurface ocean temperature distributions are more relevant.
Response:  Per reviewer's suggestion, we have replaced "SST" with "ocean state" in most places.  However, in the historical review part, since those studies really used SST for analyses, we still keep SST there.   In addition, for some discussions on the AMIP type runs, we also keep SST.

-- Clarification regarding figure 7: is this the average of the 2003 anomalies relative to the different models' climatologies, or is it the average (over all years) model T-2m and

precipitation minus the average (over all years) observations? I'm guessing the latter, given the remarkable agreement with figure 6c,d. The latter can truly be considered a bias,but the term bias was used differently elsewhere in the paper.

Response: Figure 7 shows the average (over all years) model T-2m and precipitation minus the average (over all years) observations.  In the "Section 3.2 (2) Task 2" of the revised paper (lines 362-365), we have clearly indicated that we use the "bias" for both climatology and 2003 for simplicity as was done in Pan et al (2001) and Lavers et al. (2021).

---

## Author Response (AR2)

Referee #1: Rene Orth, rene.orth@bgc-jena.mpg.de

**Anonymous during peer-review:** Yes **No**
**Anonymous in acknowledgements of published article:** Yes **No**

**Recommendation to the editor**

**1) Scientific significance**
Does the manuscript represent a substantial contribution to modelling science within the scope of this journal (substantial new concepts, ideas, or methods)?

**Excellent** Good Fair Poor

**2) Scientific quality**
Are the scientific approach and applied methods valid? Are the results discussed in an appropriate and balanced way (consideration of related work, including appropriate references)? Do the models, technical advances and/or experiments described have the potential to perform calculations leading to significant scientific results?

Excellent Good **Fair** Poor

**3) Scientific reproducibility**
To what extent is the modelling science reproducible? Is the description sufficiently complete and precise to allow reproduction of the science by fellow scientists (traceability of results)?

Excellent **Good** Fair Poor

**4) Presentation quality**
Are the scientific results and conclusions presented in a clear, concise, and well structured way (number and quality of figures/tables, appropriate use of English language)?

Excellent Good **Fair** Poor

For final publication, the manuscript should be
**accepted as is**
accepted subject to **technical corrections**
accepted subject to **minor revisions**
**reconsidered after major revisions**
  I am willing to review the revised paper.
  **I am not willing to review the revised paper.**
**rejected**

**Suggestions for revision or reasons for rejection** (will be published if the paper is accepted for final publication)
Second Review of gmd-2020-329 by Xue et al.
"Impact of Initialized Land Surface Temperature and Snowpack on Subseasonal to Seasonal Prediction Project, Phase I (LS4P-I): Organization and Experimental design"

The paper has overall improved as the authors have addressed some of the concerns raised by me and the other reviewer.

However, at the same time some important issues remain insufficiently addressed:

-- regarding main comment (1) from my previous review
I agree with most of the author's comprehensive reasoning in the rebuttal,
but these points should also be (more clearly) reflected in the manuscript.
The relevance to discuss these points is highlighted by the fact that also

Response: This is a good suggestion. In the revised version, we have
added the following sentences to lines 695-700 "LS4P focuses on
process understanding and predictability. Since the current start-of-the-
art models are unable to properly produce the observed surface
temperature anomaly and the corresponding anomaly-induced dynamic
as well as the associated physical processes in their simulations, the bias
correction in post-processing (a method that has been used for some
simulation studies) is unable to generate these processes to help our
understanding and will not be considered in the LS4P project" which was
in our previous response to comment (1).

reviewer #2 expressed concerns about the temperature masking strategy.
Related to this, I still have difficulties understanding Figure 2;
in particular it is not clear to me how the starting points of the arrows and
their directions are chosed or supposed to indicate.
The same goes for equation 1b; let's assume we have
T_obsanomaly = -2 & T_bias = +1, then
deltaT_mask = -1 (with n=1),
but actually it should be -3 as far as I understand (?)

Response:
We have further revised the figure to make it more clear and have added the following in the
Figure note:
 "4). $T_0$ is the initial condition for Task 1 and  is the initial condition after imposing the mask
for Task 3" .

The conditions in Eq. 1a and 1b, i.e., when anomaly and bias have the same sign or different
 was correct in our original manuscript.  After many iterations
between our co-authors, we actually found this error which appeared in one version of the
paper and we thought that we had corrected this error.  But somehow, the error still re-
appeared.  We are very sorry for this oversight and appreciate the reviewer's careful checking
to find the error in our equations.  This is now corrected in Equation 1.

-- regarding main comment (2) from my previous review
I appreciate that the authors have added table B1 and information on the GLASS
satellite datasets. However, I am still missing justification and information
on the CMA data which I feel is important as this is used as ground truth here.
The manuscript mentions that data from 80 stations across the Tibetian plateau
is used. Given the significant area of the plateau, this means that some kind
of interpolation in space (and time?) is required. In this context I am wondering
how this is done, and how it compares with the ERA5 modelling and data assimilation
system which probably uses less station data from that region but can efficiently
interpolate between them, also because it ingests and benefits from observations
across multiple sources and variables.
Response: According to our knowledge, the ECMWF (and probably any other
reanalyses data and observation data) only has less than 20 stations data in Tibetan
Plateau through GTS and collaborative agreement and less than 300 stations in China,

while our data sets have 80 stations over Tibetan Plateau and more than 2400 stations over China. There are considerable differences. However, it is improper for us to announce how many stations in other relevant data sets over Tibetan Plateau because no openly published resources provide such information. But we are confident that other data groups, such as CRU, CAMS, or ECMWF, know, from their experience, our data are probably the best openly published data sets for China, especially the Tibetan Plateau.

We fully agree with the reviewer that spatial interpolation over Tibetan Plateau is quite challenging. Indeed, reanalysis data benefits from non-local information being assimilated into their systems and effectively advected into our region of interest by the model dynamics (and physics interactions). But owing to the large amount of data at our disposal, we still feel that our analysis is likely superior over this region. The paper (Han et al., 2019) that is cited in the current study described the spatial interpolation and data processing methods in detail for their data set. We did not emphasize this aspect in the previous draft. In the revised version, on line 276, we added a sentence "A detailed spatial interpolation method for the data sets are discussed in Han et al. (2019). "

-- regarding main comments (4) & (5) from my previous review
I understand the authors reply. But this needs to be more reflected in the manuscript such that readers do not get confused between the description of the entire project and of phase I as one part of it. For example the title mentions phase I, so I would not be expecting to find motivation for phase II (snow, predictability) in this paper.

Response: Thanks for the reviewer's comments. Per the reviewer's last suggestion, we now greatly deg-emphasize the aerosol-in-snow effects and Phase II in the last revision. We only present Phase II at the end of the paper in this version. We also keep to a minimum description of our future work just for the reader's reference/information since when readers read this paper, it is natural for them to wonder what is next for this project. And of course, it is also a very common approach to very briefly mention future work at the end of a paper but indeed we have modified and tried to clarify this. For the aerosols-in-snow, we have only discussed this in one place. Snow processes and aerosol interactions are very hot topics with broad interests and owing to the location of our study, are very relevant to our project. If we zero out those things, we feel that some readers will immediately raise questions as to why we didn't mention them.

Figure 3:
Why is one of the maps global while the remaining maps are focusing on the Tibetian Plateau?

Response: Only one panel in the figure shows the entire globe because we want to clearly show that the anomaly only covers Tibetan Plateau and give a sense of its relative size. We revised the sentence on lines 444-446, "a mask using Equation 1b was generated and only imposed over the Tibetan Plateau region as demonstrated in the global map (see Figure 3c)." to explain the idea why we use a global map here.

Thank you very much for all the reviewer's very careful efforts.

I do not wish to remain anonymous - Rene Orth.

**Anonymous during peer-review: Yes** No

**Anonymous in acknowledgements of published article: Yes** No

**Recommendation to the editor**

**1) Scientific significance**
Does the manuscript represent a substantial contribution to modelling science within the scope of this journal (substantial new concepts, ideas, or methods)?

Excellent **Good** Fair Poor

**2) Scientific quality**
Are the scientific approach and applied methods valid? Are the results discussed in an appropriate and balanced way (consideration of related work, including appropriate references)? Do the models, technical advances and/or experiments described have the potential to perform calculations leading to significant scientific results?

Excellent **Good** Fair Poor

**3) Scientific reproducibility**
To what extent is the modelling science reproducible? Is the description sufficiently complete and precise to allow reproduction of the science by fellow scientists (traceability of results)?

Excellent **Good** Fair Poor

**4) Presentation quality**
Are the scientific results and conclusions presented in a clear, concise, and well structured way (number and quality of figures/tables, appropriate use of English language)?

Excellent **Good** Fair Poor

For final publication, the manuscript should be
**accepted as is**
accepted subject to **technical corrections**
**accepted subject to minor revisions**
reconsidered after **major revisions**
    I am willing to review the revised paper.
    I am **not** willing to review the revised paper.
**rejected**

**Suggestions for revision or reasons for rejection (will be published if the paper is accepted for final publication)**
For this re-review, I include my original review comments, the authors' responses, and my responses to their responses, when I have them. (Only a subset of the minor revisions are reproduced here.) Overall, I think that with the addition of some additional

caveats (see below) and a serious reconsideration of how the results in Figure 6 are presented (and perhaps replaced), the paper will be ready for publication. I don't need to see it again.

[Original: This paper describes the international / multi-organization LS4P project and provides some initial analyses of data submissions. It is essentially an "introduce the community to the project" paper, with most of the scientific findings to be documented later, as the project progresses.
While the paper is well written, in my opinion it has enough issues to rate a review of "major revision". On the one hand, I do applaud the experiment organizers for bringing together such a wide-ranging group of participants to address such an important problem. With this diverse a group and a corresponding collection of model outputs from such a diverse set of models, I don't doubt that the project will bear useful scientific fruit, and a paper like this that introduces the project to the broader community is certainly of value. On the other hand, the paper's write-up glosses over several critical aspects of seasonal and subseasonal prediction that call some of the project's long-term strategies into question, at least in terms of how they're currently described. A revised version should address these shortcomings through substantial qualification (not just a sentence or two here and there) or, better yet, through a substantial rethinking of the approaches to be applied.]
[Response: We thank for the reviewer's acknowledgment that we have a wide-ranging group of participants to address an important problem and that the project will bear scientific fruit. We have taken into careful considerations of the reviewer's comments/concerns on our writing and questioning of our strategies. The following is our point-by-point responses to address the issues that the reviewer raises. We believe some issues are rooted in the reviewers understanding in our experimental design, thus in our revisions, we have made a great deal of effort to improve the presentation of our experimental design section. The following responses explain and address the reviewer's questions.]

Current review: I continue to think that introducing the experiment to the community through a paper like this is valuable, and I again applaud the authors for organizing such an extensive group. Also, however, I think I understood the experimental design from the beginning, so many of my original comments are still valid. See below.

Response:  Thank you.  Please see our point-by-point responses below.

[Original: 1. The underlying assumption of the project appears to be

that if a model does not produce an accurate temperature over the Tibetan Plateau, the fault lies with the land model (e.g., in how long the land model maintains an initial condition). The paper states this explicitly on line 465 (original submission). The truth is, all models have biases in both air temperature and precipitation across the globe, and these biases could have any number of sources. A temperature bias over the Tibetan Plateau might have nothing to do with land model processes. It might instead result from deficiencies in the reproduction of the general circulation, for example, or from some problems with the radiation balance. Forcing the model to have a low temperature bias by imposing a stronger initial temperature anomaly (perhaps even an unrealistic anomaly, through eq. 1) may amount to "getting the right answer for the wrong reason", which is not a good basis for a forecast experiment. It's quite possible that forcing a correct temperature through such an initialization when the model wants to do something else for reasons unrelated to land processes might have unexpected negative consequences – especially if the model is artificially modified in one region and not in surrounding regions. Substantial discussion regarding this is needed.]

[Response: we fully agree with the reviewer's comments that "all models have biases in both air temperature and precipitation across the globe, and these biases could have any number of sources," and when you correct the bias you may be "getting the right answer for the wrong reason".

But we have to clarify a few issues:

(1). Numerous modeling studies, since the very beginning of meteorological model development, have worked on correcting model-produced precipitation and temperature errors by improving some model parameterizations and through improving initial and boundary conditions. Our approaches and statements are based on a number of published papers in our field's major journals. We cannot simply speculate these peer reviewed research's results are produced due to the wrong reason unless there is an evidence to support such statement. Otherwise, we may eliminate any modeling improvement studies because, at least at its early stage, they cannot prove that improvement is not due to a wrong reason. Only the community collective efforts with long term exercise can prove it. ]

Current review: That's fine, but you're doing the opposite here, which is just as bad – you are not warning the reader, if only with a simple caveat, of the possibility that you are correcting the wrong source of the error. In addition, simply imposing a large temperature

anomaly (possibly even larger than the observational anomaly) couldn't really be called an improvement of the model.

Response: We understand the Reviewer's concern, which actually is also our concern. Our data group has, through a long time effort, recently produced the daily Tibetan Plateau surface temperature dataset. In the next LS4P publication in a LS4P special issue, we will show that our model imposed and simulated surface temperature anomalies are actually smaller than the observed anomaly in early May. We have revised the sentence on lines 561 to 564 as following "In the LS4P-I experiment most models are only able to partially produce the observed T-2m anomaly in May despite the imposed initial masks. The recently available daily Tibetan Plateau surface data from the LS4P data group show that our imposed initial anomaly is not extreme, but the models lost the imposed anomaly rather quickly."

[More response: (2). The reviewer speculates the problem in simulating the LST anomaly may not be due to the surface models. In general, when we try to correct a model deficiency in one variable, it is normal to check the dynamic and physical processes relevant to this variable first. For instance, for the precipitation simulation errors, we naturally check the convective and cloud process modeling and surface evaporation parametrization first. Unless there is an evidence to show these are other reasons, we cannot claim it is a wrong approach.]

Current review: True, but again, you should at least acknowledge, with a caveat, that it *may* be a wrong approach. Based on the text, an uninformed reader will assume that it is common knowledge that land processes are unquestionably responsible for the bias.

Response: In the revised version, on lines 700-701, we point out that "we encourage/welcome different approaches to tackle this issue, and for comparison with the approach presented in this study".

[More response: Our statement on land model deficiencies are based on published papers and analyses from the LS4P research. On lines 126-129, we present a publication (Liu et al., 2020), which focuses on exploring the causes of model deficiency in properly producing the observed surface temperature anomaly in high mountains. That study demonstrated this deficiency IS associated with the land surface process model, including snow/albedo and soil subsurface memory effects. Recently, we had another published paper (Li et al., 2021) address this issue and further confirms Liu et al.'s conclusion.]

Current review: I'll admit to not having done a separate review of these papers, given time constraints. However, looking at them quickly, they appear to focus on a single modeling system and shouldn't be assumed to represent models in general.

Response:
Yes, indeed the studies refer to single-model results.  But the research in our field always starts from a single model (as a proof of concept in a sense), and then follows by many single model studies as other groups investigate what they see as an idea with potential merit (or not).  Most published modeling papers are single-model studies.  The scientific value of a research question is based on the dynamic and physical principles presented in the study.  These single-model research studies at least demonstrate a possibility and show it may be worth to pursue more research on that direction.  A community effort, such as LS4P, always needs such research to motivate the participants and as justification since such community efforts are very time and resource consuming.

[More response: In addition, we have also analyzed the reanalysis data, which are used for model initialization, and indicated that the deficiencies in reanalysis data in high mountain areas also contributes to the simulated surface temperature bias. We believe we have caught the main (if not all of the) reasons for the model deficiency for this aspect. We have more clearly indicated our statements are based on the published papers in abstract and conclusion, and welcome more research to explore this issue in the revised version (Lines 493-496 )]

Current review: All I'm saying is the text reads as being highly certain that land processes (and reanalysis data) are responsible for the model biases seen across the models. This, I feel, needs to be tempered with at least some explicit admission – a simple, explicitly stated caveat to the reader – that modifying the land states *may* be reducing temperature biases for the wrong reason.

Response:  In the revised paper, on lines 501-502, we point out that "Further development is necessary to improve this approach." On lines 700-701, we also point out that "we encourage/welcome different approaches to tackle this issue, and for comparison with the approach presented in this study."

[More response: (3). However, to improve land model and reanalyses data for this aspect are not simple tasks and may take decadal effort (today's land temperature model development has a more than 70-year history), but proper S2S prediction, including drought/flood/heat waveprediction, is a WMO task requiring urgent attention owing to a significant societal demand. On line 377-379, based on the LS4P modeling group's practice, we have pointed out that preliminary research suggests "prescribing both LST and SUBT initial anomalies based on the observed T-2m anomaly and model bias is the only way for the current ESMs to accurately produce the observed May T-2m anomalies". In fact, using initialization to

improve meteorological prediction is nothing new but a traditional meteorological approach. We believe when this issue gets more attention and more data are/become available, more methodologies may be developed to address this issue. Butthe scientific development always takes time. We have to allow any development to takestep-by-step improvement.
(4). We agree with the reviewer that the statement on line 465(previous version) may be misleading and therefore we have modified the text to more properly reflect the ideas that we present in this paper (Lines 493-501).]

[Original review: 2. The overall strategy seems to ignore the fact that forecast models generate their forecasts relative to their own climatologies. A model that is known to be biased warm may produce an anomalously cold 2003 over the Tibetan Plateau (compared to what it usually produces there), and that would be useful information even if the forecasted temperatures are still warmer than the average observed TP temperature. The point is that people know how to account for long-term model biases. They would properly consider a forecast model's result to be "2003 will be colder than usual by 5 degrees" rather than "the temperature will be 20C". The emphasis here on matching the observed temperature in absolute magnitude seems inappropriate to a discussion of forecast systems. See, e.g., the NMME forecast anomaly pages [https://www.cpc.ncep.noaa.gov/products/NMME/seasanom.shtml], which show forecasted anomalies relative to each model's climatology.]
[Response: The reviewer raises several issues here.
(1). Because of the model systematic bias, some groups indeed applied the anomaly prediction in their normal practice. We understand the justification for these groups, including some LS4P groups, used the anomaly predictions, but predicting the temperature in the real world is always our ultimate goal because the public needs the forecast for the real world, not the forecast relative to one group's model climatology. ]

Current review: I think the authors are missing the point of my comment. The public needs a useful forecast, and if it can be effectively produced by bias-correction after the forecast is made, so be it. Anyway, no additional response is needed here.

Response:  Thanks.

[More response: (2). However, in the LS4P experimental design, as the first step, we only intend to see if there is any relationship between the observed Tibetan Plateau spring LST/SUBT anomaly and downstream summer precipitation anomalies. We mainly look at

the difference between Task 3 and Task1. In this way, the model systematic bias has been eliminated so what we really look is indeed the anomaly, which is just what the reviewer tries to emphasize here.]

Current review: Yes, that sounds good.

Response:  Thanks.

[More response: Because of this approach, our goal here does not emphasize producing the best initialization for May 2003 per say, although the methodology present here should serve this purpose. We have made major revision for Section 3.2 (3). In the modified manuscript, we have clarified our idea/approach. We acknowledge that our schematic diagram and relevant text do not emphasize this idea clearly as pointed out in the Reviewer's third minor comment. We have modified the schematic diagram (Figure 2), figure captions, and reorganize the "Section 3.2(3) Task 3" in the revised manuscript to make the idea clearer. Original Figure 2 includes both warm and cold years. In the revised version, to make the thing simple and less confusion for readers who are unfamiliar with LS4P, we only include cold year (same as the case in the LS4P Phase I) in the text and move the schematic diagram for the warm case to appendix for readers who want to pursue this issue further for their own research. In our response to the reviewer' 3rd minor comment below, we will have some more detailed explanation. ]

[Original review: 3. Forecast systems also produce a range of forecasted values through the running of ensembles, and any one ensemble member could represent what happens in nature. The experimental analysis protocol, however, emphasizes the importance of having the *ensemble mean* match the observed anomaly. This is inappropriate. The key question is, do any of the ensemble members look like the observations? (And, in conjunction with point #2 above, the truly key question is, do any of the ensemble member anomalies *relative to the forecast model's climatology* look like the observed anomaly?) A model cannot be considered wrong if one of its ensemble members looks like the observations. Insisting that an ensemble mean match a specific year's temperature seems wrong.]

[Response: In this paper, we do not emphasize the model intercomparison as well as each model's evaluation because the major focus for the LS4P is whether the LST /SUBT can provide S2S predictability through the multi-model efforts, which idea has never been tested before. Since many multi-model projects, such as CMIP, WAMME, and many others, find the ensemble means normally produce better results than any individual model's performance, we use the ensemble mean and the range of individual model results to

assess whether there is S2S predictability and its uncertainty. The reviewer may have different opinion on this, but this is a common approach currently used in multi-model studies in the community, such as CMIP, AMIP, WAMME and endorsed by the LS4P modeling groups. In addition, the LS4P has more than 20 ESMs and many RCMs. A comprehensive analysis of each model performance is not that closely related to our main focus at this stage.]

Current review: The authors have missed my point here. I don't doubt that multi-model forecasts can be better, but that has nothing to do with it. My point is that what happened in the real world in 2003 is one of multiple possible "ensemble members" of the real world system. The models can actually provide a range of what might happen through their ensemble spreads. If any given model's ensemble spread captures the single-ensemble-member representation of nature, it can't be called an error., and it needn't be considered something to correct. (This goes for the 2003 focus considered for most models. For biases that appear over multiple years, then I agree, there is an issue.) Looking at ensemble means – and insisting that the ensemble means match the single ensemble member of nature rather than capturing it within its intraensemble range – is what I took issue with. I was asking for some discussion of this in the paper.

Response: Thanks for the reviewer's clarification.  We agree with the reviewer's opinions that 2003 is only a single ensemble member in the real world.   We also agree that "For biases that appear over multiple years, then there is an issue.".

In the paper, we have Figure 7 to show the biases that appear in 2003 actually also exist in these models' multi-years' climatological simulations.  As such, we demonstrate the error shown in the 2003 case is an issue and correcting these biases has a broad implication.

The reviewer is roughly correct in noting, "If any given model's ensemble spread captures the single-ensemble-member representation of nature, it can't be called an error, and it needn't be considered something to correct." For an agreed on index or spatial pattern, in principle, probabilistic statements could be made about likelihood of an observed value being drawn from the given ensemble. In future, we could include information from all ensemble members to permit such analysis. However, at this point, the aim is to establish some basic features across the multi-model ensemble.  We have included a sentence noting the reviewers caveat on lines 635-638, "We note the caveat that the ESM results are from ensemble means, and in comparing to a particular year the spread of the ensemble results is also important. But one can immediately see that the biases are substantial, despite the particular combination of ESM results indexed to the Tibetan plateau temperature". Furthermore the reviewer's point about the influence of the climatological bias is indeed relevant and we have revised the sentence on lines 662-663 regarding figure 7 to clarify this, since this was one of the main reasons for including figure 7 in the first place.

We agree that we cannot determine a single model is correct or incorrect just based

on one case since single model can get the right answer by chance. That's why we use an approach with ensemble integrations by multiple models, because it reduces systematic bias of each model. Furthermore, we also need ensemble observations over many cases, in order to estimate the spread of the observations. The case example of 2003 is just a starting point for LS4P, indicating that we are in the right direction. We need more cases to show that the observational spread is captured within the ensemble model spread.

At this stage, every weather forecast center still uses "equitable threat scores and bias (same as what we did)" to assess a prediction. The reviewers' comments are "philosophical" and imply a new way to assess the prediction. It is hard for us to make a statement or even a hypothesis for the reviewer's approach at this stage without more information/evidence. By the way, for the LS4P, the main goal is to test the hypothesis described in the paper. Assessing each model's performance is not our focus.

But one thing is for sure: eventually we will need more cases to finally confirm the hypothesis. In the revised paper, on lines 720-721, we added the following statement that "Phase I focuses on the Case 2003. In the ensuing LS4P activity, more cases will be tackled, which will further improve our assessment on the ESM's predictability linked to LST/SUBT."

[Original review: 4. The model results concerning May Tibetan Plateau temperature anomalies and June precipitation anomalies in east Asia is perhaps suggestive but far from indicative of a causal relationship. Even if the agreements in 6a/6c and 6b/6d do suggestthat one pattern led to the other (it could very well be coincidental),I don't see how the Tibetan Plateau in 6a/6c can be isolated as the source of the agreement in 6b/6d. Significant qualification of this figure's implications is needed.]

[Response. We fully agree with the reviewer's comments and apologize not to have presented our ideas more clearly. Figures 6a/6c and 6b/6d are NOT intended to present the causal relationship. These figures only intend to explain how we develop our hypothesis. Observational results in Figure 6A, B (from various observational data sets) along with the remarkable consistency of modeling results in Figure 6C, D (compared to Fig. 6A, B) together provided the underpinnings for the LS4P conjecture that if the May land surface temperature anomaly on the Tibetan Plateau does contribute to the June East Asian precipitation anomaly, then improving the May land surface temperature simulation over the Tibetan Plateau through an improved initialization should allow Earth System models to better predict June East Asian precipitation.]

Current review: With my above points about looking at ensemble means in mind, I want to raise a very important issue about Figure 6. In their response above, the authors point to "the remarkable consistency of modeling results in Figure 6C,D (compared to Fig. 6A, B)." In studying the figure, I've come to the conclusion that the agreement is probably not nearly as remarkable as the authors think. As I understand it, Figure 6b shows [2003 observed

precipitation] minus [long-term observed climatology]. Because most of the models (11 out of 13) had warm biases, the differences shown in Figure 6d mostly represent [2003 observed precipitation] minus [ensemble mean 2003 precipitation]. The point is that the action of taking an ensemble mean brings the signal (the models' ensemble mean 2003 precipitation) much closer to their own climatologies, so that the differences in Figure 6d, just like those in Figure 6b, *mostly* reflect the observed anomaly. Presentation of Figure 6 without this discussion would be misleading to the reader. Highlighting the boxed regions in the figure would be inappropriate; nature just happened to dump a lot of rainfall in the top box and relatively little in the bottom box, and that's what dominates the differences shown in these boxes in both 6b and 6d.

This said, I'm intrigued by the statement on line 637: "The models with opposite sign of T-2m bias produced the opposite precipitation response." This may indeed be relevant and would be worthy of a figure (one panel showing precipitation biases for the cold T-2m bias models, and the other panel showing the biases for the warm T-2m bias models, without an imposed sign change). This, I feel, would have greater relevance than the current Figure 6. I would be impressed, for example, if the precipitation anomalies in the boxed regions were indeed reversed between the two bias cases.

Response:  For figure 6, we have stated the statistical correlation coefficient in the text, which is 0.62.  For a climate study, it is a very high correlation.  In most SST studies, the correlation is about 0.4.  The word "remarkable" is used in our private discussion and has not been used in the text.

The reviewer's understanding of Figure 6c is incorrect.  In the figure subtitle, we indicate that it is the ensemble mean's bias, i.e., [ensemble mean 2003 precipitation] – [2003 observed precipitation], not [2003 observed precipitation] minus [ensemble mean 2003 precipitation] as the reviewer indicated above.  If the reviewer's hypothesis is correct, i.e., ensemble mean is close to model climatology, then the difference in Figure 6c should multiplied by (-1) to be consistent with Figure 6c.  But this is not the case.

Furthermore, we disagree with the reviewer statement: "action of taking an ensemble mean brings the signal (the models' ensemble mean 2003 precipitation) much closer to their own climatologies". If the model ensemble mean for a specific year only represents the model's long term climatology, how could the ensemble mean to use in the prediction to produce different year's climate variability?

As for the suggestion of separating warm models and cold models, it is a very good suggestion. However, there is only two cold bias models. The results from two models lack statistical significance and cannot support any conclusion. In the further studies, with more cases available, we will address this issue. We delete the original line 637 "The models with opposite sign of T-2m bias produced the opposite precipitation response".

As the boxes in Figure 6, it is only for readers who are unfamiliar with the East Asian Geography to understand the geographic names and locations that we present on Page 13 (a footnote was added on that page to refer to Figure 6). The box is not relevant to the reviewer's comment. The reviewer's comments here show the reviewer may have suspicions about the climate predictability. We respect the reviewer's opinion. But the LS4P's efforts is to prove the predictability though a community effort.

[More response: The scientific development normally starts from scientists' curiosity based on some preliminary discoveries. The reviewer apparently is an expert in the Earth system modeling and should understand such consistency in Figure 6 from various observational data sets and various Earth system models (with very different dynamic processes and physical parameterizations) is not by chance, and is worth to propose a hypothesis then explore this issue further. By and large, Figure 6 is an important step to motivate the hypothesis. Only Task 3 (with Task 1) is designed to prove the casual relationship. We have more explicitly emphasize this point in the revised paper to avoid confusion (Lines 349-353, 624-625).]

Minor comments:

[Original review: -- I studied Figure 2 for a long time and still can't make sense of it. Why, for a cold year initialization, does the initial condition for a model with a cold bias get set to climatology whereas that with a warm bias does not? Also, please clarify in the caption: are the biases discussed here errors for the particular year of simulation, or are they long term climatological biases? I'm guessing the former, since Task 2 would need to be done for the latter; in that case, though, the use of the term "bias" is confusing here. Bias should refer to a long-term climatological error (reflecting a model deficiency), not to the error at a specific time (which should reflect both bias and random error). Overall, Figure 2 is not helpful for explaining the approach. And again, based on my earlier comments, I'm not convinced the Tmask strategy is appropriate anyway.]
[Response: This paper is for the LS4P experimental design and Figure 2 shows how we tried to generate the observed T-2m anomalies using the imposed mask in initialization then checked

whether this anomaly improves the S2S predictability and leads to better prediction of the observed drought and flood events. Based on the reviewer's comments, we recognize that we did not explain the idea clearly in the previous figure, figure caption, and text.

Although the researchers (co-authors of this paper) working in the LS4P know the idea from Figure, but probably not the readers who are unfamiliar with the project. We have performed several iterations within co-authors to improve the figure, figure caption, and presentation in the text. The section 3.2(3) is reorganized. We hope that the revised figure and text clear-up the confusion.]
In the revised figure, we have clarified which initial temperature is for Task 1 and which is for Task 3 with more explanations and indicate that the LS4P phase I's goal is to prove the causal relationship between the Tibetan Plateau spring LST/SUBT anomaly and large-scale summer precipitation anomaly. Figure 2 and Equation 1 show how to produce observed surface temperature anomaly through Task 3 initialization (relative to Task 1's initial condition). We believe these should help readers to understand the ideas better in this figure. With the revised figure, the readers can see that when we use the difference between Task 3 and Task 1, we actually try to avoid the model systematic bias which was precisely suggested by the Reviewer in the main comments.
Regarding the bias, here we did not clearly indicate whether this should be a specific year or a climatology because it depends on case and data availability. "Bias" is not always associated with climatology. As Pan et al (2001, JGR) state in their analysis that "Both GCM and RCM fields can exhibit substantial systematic differences from gridded observational data. Such discrepancies between simulated and observed fields are commonly referred to as biases", although "some differences clearly are not biases in the strict sense, but for simplicity we use the term "bias" to refer to the entire set of comparisons". Such interchanges are also used in other studies/fields. A recent paper "Precipitation Biases in the ECMWF Integrated Forecasting System" by Lavers et al (2021) discusses using the IFS control forecast from 12 June 2019 to 11 June 2020 to show that in each of the boreal winter and summer half years, the IFS has an average global wet bias". The way they use bias is similar to what we use. In remote sensing, "bias correction" terminology is also commonly used. Moreover, as discussed in the paper, from Task 2, we know the climatological T-2m bias and year 2003 T-2m bias are very consistent. Therefore, we point out this terminology issue but still keep "bias' in our paper for simplicity as did in other current practices on lines 362-365.]

Current review: Just so the authors know, Figure 2 is still confusing. It appears to suggest that models with a cold bias should set their initial temperatures to the observed climatology. I don't think that's

what they mean to say.

Response: The project intends to produce the observed anomaly from Task 3 minus Task 1 to test our hypothesis. Based on this objective and Equation 1, ideally, after imposing the mask the model with cold bias should set the initial temperature to the observed climatology as shown in Figure 2a. However, due to the tuning parameter "n", it actually would not be the observed climatology.

[Original review: -- Equation 1 appears to be a means to impose an artificially large temperature anomaly at the start of a simulation so that the anomaly is maintained realistically during the forecast. As far as I can see, there's no physical basis for the equation; it's fully empirical and could lead to initial temperatures that make little physical sense (e.g., colder than the model ever gets). More qualification is needed regarding how artificial this construct is. (And again, based on my earlier comments, it may not be appropriate to fix the temperature error in this way, since it may have a source other than the land model.)]

[Response: The LST/SUBT approach is a new development and is at a very early stage. The importance of memory of surface temperature has still not been fully recognized by the community. Currently, no ESMs, including reanalyses, are capable to reproduce observed high mountain T-2m anomaly. Developing adequate numerical methods and physical parameterizations to permanently solve the issue may take decades or longer.

In meteorology, using the initialization scheme before we develop better models and better data sets to improve the prediction is a very common approach, as done by Yeh et al. (1984, MWR) and Yang et al (1994 MWR). Those initialization strategies were always based on some empirical relations and not a strict physically-based approach. Especially in the early stages, some approaches are highly idealized. For instance, in Koster at al. (2004), which is a rather famous study, were used an approach for which a soil moisture value is artificially imposed for every time step. But that did not prevent that paper from receiving more than 2250 citations and from becoming a classical paper in meteorology. On other hand, our approach is not that extreme. The reviewer thought we may impose an artificially large forcing because of a tuning parameter. In fact, this is not the case. In the follow-up paper in a Climate Dynamics special issue, we will show every model's imposed forcing. in fact, they are not extreme. The LS4P includes most of major climate centers in the world. If our approach is totally different from their normal practices, they would not endorse the LS4P and participate in this project. If we wait until the best dynamic and physical method are developed, nothing will happen.]

**Current review**: If the authors are finding that the tuning parameter is not, in fact, extreme, they should state this here, referring to the other study. As the text now reads, the reader has no basis for thinking that the correction won't be unrealistically large. In fact, lines 556-557 are a little troubling: "…despite the fact that the imposed initial masks normally contain much larger anomalies thanthose observed". Does this contradict the above statement that every model's imposed forcing is not extreme?

Response: In the response to question 1, we have pointed out that the imposed anomaly is not extreme based on the recently available daily data from observations. We have revised the sentence on lines 561 to 564 as follows: "In the LS4P-I experiment, most models are only able to partially produce the observed T-2m anomaly in May despite the imposed initial masks. The recently available daily Tibetan Plateau surface data from the LS4P data group show that our imposed initial anomaly is not extreme, but models lost the imposed anomaly rather quickly."